# Three-dimensional and linearized mapping of vibrissa follicle afferents

Ben Gerhardt [1], Jette Alfken[2], Jakob Reichmann [2], Tim Salditt [2,4] & Michael Brecht [1,3,4] ✉

Understanding vibrissal transduction has advanced by serial sectioning and identified afferent recordings, but afferent mapping onto the complex, encapsulated follicle remains unclear. Here, we reveal male rat C2 vibrissa follicle innervation through synchrotron X-ray phase contrast tomograms. Morphological analysis identified 5% superficial, ~32 % unmyelinated and 63% myelinated deep vibrissal nerve axons. Myelinated afferents consist of each one third Merkel and club-like, and one sixth Ruffini-like and lanceolate endings. Unsupervised clustering of afferent properties aligns with classic morphological categories and revealed previously unrecognized club-like afferent subtypes distinct in axon diameter and Ranvier internode distance. Myelination and axon diameters indicate a proximal-to-distal axon-velocity gradient along the follicle. Axons innervate preferentially dorso-caudally to the vibrissa, presumably to sample contacts from vibrissa protraction. Afferents organize in axon-arms innervating discrete angular territories. The radial axon-arm arrangement around the vibrissa maps into a linear representation of axon-arm bands in the nerve. Such follicle linearization presumably instructs downstream linear brainstem barrelettes. Synchrotron imaging provides a synopsis of afferents and mechanotransductory machinery.

Facial vibrissae are important tactile sensors that likely played a key role in the evolutionary success of mammals[1]. Small rodents rely heavily on vibrissae for navigating[2] and behaviors ranging from social touch[3] to wind-sensing[4,5] and texture discrimination[6]. Rodent vibrissal sensing is an active process, in which rhythmic back and forth movements (whisking)[7,8] scan the environment, similar to blind localizing objects by scanning the floor with a stick[9]. Specifically, rats protract whiskers in precisely controlled 3D kinetics[10] to contact objects with as many whiskers as possible and as lightly as possible (minimal impingement)[11].

Object contacts deflect the vibrissal shaft and deform mechanosensory afferents densely populating the vibrissa follicle[12]. Afferent responses are tuned to the angular displacement of the shaft[13,14], and respond with astounding temporal precision[15,16], providing information about object distance[17] and orientation[9,18,19]. Shaft deflections act upon the vibrissa with characteristic angular and curvature variables and thus encode pre-neuronal information of object touch locations[20]. Seminal work has identified how receptor location contributes to response properties by combining intra-axonal recordings of identified afference types and ultrastructural analysis[21].

Anatomical studies described the intricate architecture of the vibrissa follicle sinus complex[22,23], an encapsulated structure with several blood-filled chambers. In rodents, most afferents end below or at the ring-sinus level (one such blood-filled chamber), which contains the ringwulst. The C-shaped ringwulst, attached to the mesenchymal sheath, reminds of a neck-pillow that floats between the blood

[1]Bernstein Center for Computational Neuroscience Berlin, Humboldt-Universität zu Berlin, Berlin, Germany. [2]Institut für Röntgenphysik, Universität Göttingen, Göttingen, Germany. [3]NeuroCure Cluster of Excellence, Humboldt-Universität zu Berlin, Berlin, Germany. [4]These authors jointly supervised this work: Tim Salditt, Michael Brecht. ✉e-mail: michael.brecht@bccn-berlin.de

chambers of the ring sinus and cavernous sinus, constricting afferents around the vibrissal shaft. The mechanosensory supportive machinery likely plays a decisive role in vibrissal transduction. Optical inaccessibility and the difficulty of reconstructing hundreds of afferents through serial sections prevented resolving the three-dimensional architecture of the afferent population so far. Here, we employed synchrotron powered X-ray phase contrast tomography to resolve vibrissa follicle structure by short wavelength photons that readily penetrate large bodies. Phase contrast imaging exploits the phase signal of X-rays[24] and provides orders of magnitudes more signal than absorption-based contrast alone[25] for low contrast biological tissue. A large field of view (>1.5 mm), fast image acquisition (3 min per volume) and high resolution (0.65 μm isotropic voxel size) add to the power of such imaging. In our configuration X-ray phase contrast tomography allows relatively complete, dense reconstruction of myelinated axons and provides partial information about non-myelinated axon parts. We ask about afferent types, the mapping of afferents onto the vibrissa follicle sinus complex and the insertions of afferents into the vibrissal nerve.

## Results

### Dense reconstruction of vibrissa follicle innervation

We imaged three extracted (one osmium-stained, one (partial) osmium and uranyl acetate stained and one unstained) paraffin-embedded rat C2 vibrissa follicles (Figs. 1a and S1) at the GINIX

parallel beam setup (DESY, Hamburg) (Fig. 1b). After phase retrieval, tomographic reconstruction and stitching, our image volumes (stained: 0.9 × 0.8 × 2.6 mm, stained (partial): 0.9 × 0.9 × 1.4 mm and unstained: 0.9 × 1 × 2.3 mm) each spanned the whole follicle at 0.65 μm isotropic voxel size (Fig. 1c), except the unstained (partial) dataset, which did not cover the distal follicle completely (Fig. S1). In virtual 2D sections (Fig. 1d) myelinated axons can be recognized (Fig. 1d, inset) along with major follicle structures. While in our stained dataset myelin sheaths are readily visible, axons become visible as dots in the unstained dataset, which led to two different approaches of reconstructing myelin sheaths of myelinated and stained axons, and the lumen for myelinated unstained axons; both approaches revealed a similar and highly stereotyped picture of axonal architecture (Fig. S1). In the remainder of the manuscript, we focus on the analysis of the stained and complete dataset, which allowed for a more comprehensive analysis due to good axon visibility. The dataset allows the synopsis of afferent innervation and the three-dimensional follicle architecture comprising ring sinus, ringwulst, hair shaft, inner conical body, and root sheath (Fig. 1e). Our dense neuronal reconstruction emerges as a highly ordered axon-goblet inserted into the accessorial mechanosensory machinery (Fig. 1f). We conclude that synchrotron imaging reveals the three-dimensional architecture of both neural structure and accessorial mechanosensory elements (Fig. 1e, f), prerequisites for understanding vibrissal mechanotransduction.

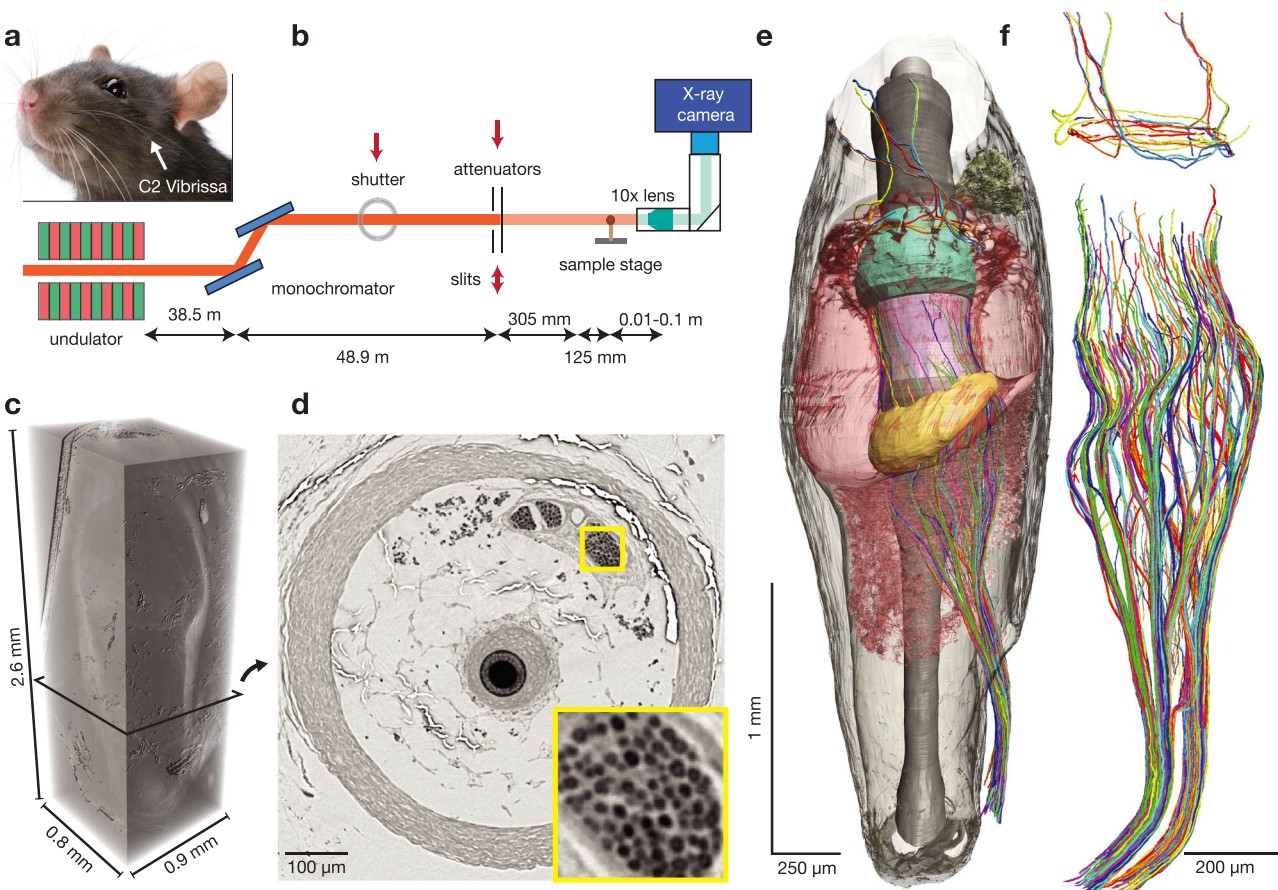

**Fig. 1 | Dense vibrissa follicle reconstruction. a** Rat face photograph showing characteristic mystacial vibrissae. **b** Schematic of X-ray phase contrast imaging at the GINIX parallel beam setup (DESY, voxel size of 650 nm). **c** 3D rendering of the obtained C2 follicle dataset. **d** Virtual 2D section through the follicle shows detailed anatomical structures (yellow inset = axonal innervation). Inset shows myelinated axons of the deep vibrissal nerve at high magnification. **e** 3D rendering of densely reconstructed follicle anatomy, including capsule (transparent grey), cavernous sinus (red), ring sinus (red), ringwulst (yellow), Merkel cell region (pink), inner conical body (mint), sebaceous gland (green) and axonal innervation (random color assignment). **f** High magnification 3D rendering of 174 myelinated and ≥58 unmyelinated deep vibrissal nerve axons (lower) and 14 circumferential lanceolate axons (upper) supplied by the superficial vibrissal nerve.

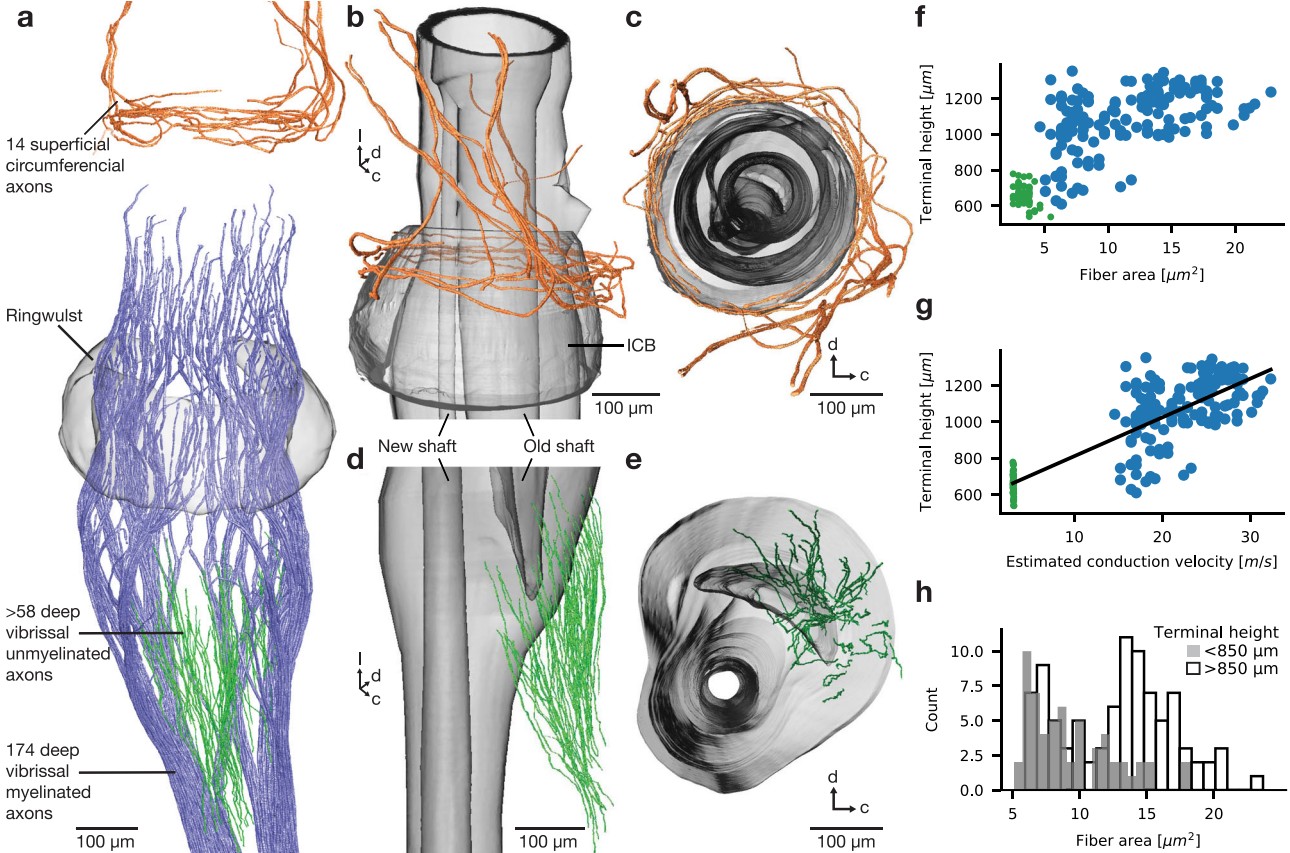

**Fig. 2 | Three follicle axon classes and axonal velocity gradient of deep vibrissal afferents. a** Myelinated (blue) and unmyelinated (green) innervation supplied by the deep vibrissal nerve and circumferential innervation (orange) supplied by the superficial nerve. Ringwulst shown in transparent grey. **b** Circumferential innervation supplied by the superficial nerve wraps around the upper region of the inner conical body (ICB, transparent grey). **c** 14 superficial axons enter in four axonal fascicles. **d** Unmyelinated innervation supplied by the deep vibrissal nerve targets the root sheath (transparent grey) in straight vertical trajectories. **e** >58 unmyelinated axons (lower bound estimate) are polarized to the dorso-caudal circumference of the vibrissa shaft (see also Fig. 3). **f** Fiber area plotted against terminal height (relative to nerve entrance into the follicle) show a gradient of increasing axon thickness. Myelinated data is shown in blue and unmyelinated in green. **g** Estimated axonal conduction velocity plotted against terminal height. Conduction velocity was estimated according to Waxmann and Bennett[28] ($r = 0.93$, $p = 1.67 \times 10^{-50}$; refers to linear regression). **h** Fiber diameter of afferents terminating below 850 μm (transparent grey) and above 850 μm (white with black outlines) relative to the nerve entrance into the follicle (see Fig. 1e). l lateral, d dorsal, c caudal.

## Follicle axon classes and bimodal diameter distribution of deep vibrissal afferents

We observed three classes of axons innervating the follicle (Fig. 2a). Distally, close to the skin, we reconstructed 14 myelinated axons with lanceolate endings, which are supplied by four superficial vibrissal nerves (Fig. 2b), and had distinct circumferential trajectories around the inner conical body (Fig. 2b, c). The bulk of the follicle innervation were 174 myelinated axons supplied by the deep vibrissal nerve (Fig. 2a, blue). In addition, we traced 58 unmyelinated axons (Fig. 2a, green; Fig. 2d), which innervate the caudal circumference of the vibrissal shaft (Fig. 2e). We note that our reconstruction of unmyelinated axons was hampered by resolution limitations. Unmyelinated axon reconstructions are therefore partially incomplete and the data referring to unmyelinated axons need to be interpreted with caution. Collectively, our tracing revealed a total of >32 cm reconstructed myelinated axonal wiring. The number of 254 deep vibrissal nerve axons corresponds closely to the ~200 myelinated and ~100 unmyelinated sensory fibers that we expected to find[26]. 3D renderings of densely reconstructed afferents (Fig. 2a) show distinct characteristics: (i) afferents ascend in ordered, straight trajectories, (ii) vertical trajectories predominate (Fig. S2), (iii) there is little branching and if so, branching is restricted to distal regions, (iv) axons ascend in arms with following distinctive polar coordinates. Such axon characteristics result in a high degree of angular specificity and confirm previous description of vertically ordered innervation trajectories[27]. We note that axon-arms were most obvious below the ringwulst. Above this level, axon-arms were still partially visible but there was no consistent arm bifurcation pattern, and we lack a stringent definition of axon-arms. Most interestingly, axonal diameters of deep vibrissal nerve afferents differed systematically as a function of the afferent ending position from the proximal to the distal follicle (Fig. 2f). Unmyelinated and thin myelinated axons innervate proximally, the thickest axons innervate distally, indicating a proximal-to-distal temporal precision gradient. We computed estimated axon conduction velocities from diameters of myelinated and unmyelinated fibers[28] along the proximal to distal axis and observed a strong and highly significant correlation (Fig. 2g, $r^2 = 0.83$, $p < 10^{-10}$). Further, we noted that this relationship of fiber diameter and terminal height follows a bimodal distribution, where most afferents terminating below 850 μm above nerve entrance into the follicle are small caliber and the majority of afferents terminating beyond 850 μm above nerve entrance into the follicle are large caliber axons (Fig. 2h). We conclude that deep vibrissal nerve afferents innervate with great angular specificity and show a proximal-to-distal axonal conduction velocity dependency.

## Types and endings of myelinated deep vibrissal nerve afferents

Next, we focus on the myelinated axons supplied by the deep vibrissal nerve. We distinguish different types of afferents building on seminal

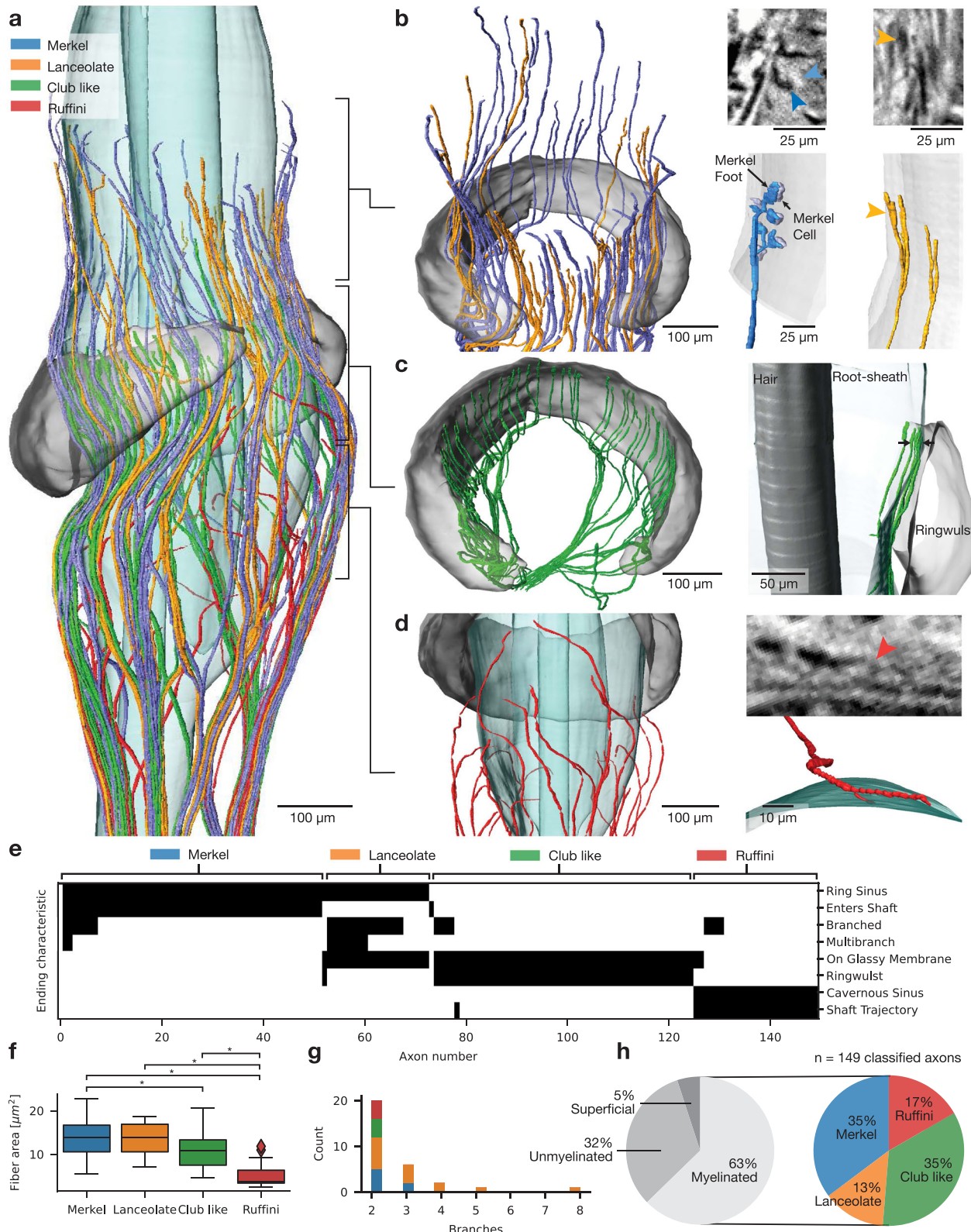

work, which disentangled sensory ending morphology and response characteristics into Merkel, lanceolate, club-like and Ruffini-like endings[19,21–23,29,30]. We classified afferents by vertical ending territories and morphology but could not completely resolve the thinnest unmyelinated parts of afferent endings. By analyzing afferents in their entirety and along with support structures, afferent types are distinguishable with great certainty as they differ along multiple dimensions. In Fig. 3a we provide a volume rendering of afferents colored

according to the four identified afferent types ($n = 149$ classified axons). Ring sinus level afferents surpass the ringwulst by 100–300 μm and separate into Merkel ($n = 51$) and lanceolate afferents ($n = 21$). Merkel afferents are distinct by piercing of the glassy membrane, distal terminal positions and even radial tiling (Fig. 3b). Lanceolate afferents terminate outside the glassy membrane in spear-like endings, show branching into 2–7 ending branches and are polarized to the ringwulst aperture (Fig. 3b). A high-resolution analysis of a Merkel and lanceolate

**Fig. 3 | Myelinated afferent types and endings. a** Whole mount myelinated axonal innervation ($n$ = 149 axons) colored by afferent type (see color legend) and ringwulst, shaft and outer root sheath in transparent grey. **b** Left: Ring sinus level Merkel ($n$ = 51) and lanceolate afferents ($n$ = 21). Merkel afferents are distinct by a kink at the axon terminal from piercing the glassy membrane (white arrow). Right: High magnification of a completely reconstructed Merkel afferent including Merkel feet (MF, blue) and Merkel cells (MC, light blue) terminating behind the glassy membrane and lanceolate ending terminating outside the glassy membrane. Virtual 2D sections of the respective endings are shown above. **c** Left: Ringwulst level club-like afferent ($n$ = 52) population. Right: Side view of club-like afferents terminating on the glassy membrane of the root sheath directly adjacent to the ringwulst. White arrows indicate pinching of the afferent terminals upon shaft deflection. **d** Left: Cavernous sinus level Ruffini-like afferents ($n$ = 25). Right: Top view of a single Ruffini-like afferent terminating on the glassy membrane below the ringwulst. Virtual 2D section of the ending is shown above. Ruffini-like endings are distinct by non-straight trajectories targeting the root sheath. **e** Ending characteristic overview of 149 classified afferents. Black ticks indicate the presence and white ticks indicate the absence of the morphological feature as displayed on the left that were scored for each afferent. Twenty two afferents remain unclassified due to incompleteness. Multibranch refers to >3 branches and shaft trajectory to almost horizontal ending trajectories (as in 4 d). **f** Fiber area (displayed as median) of classified afferents. Asterix' indicate significant differences (Kruskal Wallis: $H$ = 38.85, $p$ = 1.8 × 10$^{-8}$; Significance according to Dunn's Bonferroni corrected pairwise comparison with alpha = 0.05). Boxplots display the 25th to 75th percentile range as the box and the median as center line. Boxplot whiskers extend by the inter quartile range. Outliers are plotted individually. Note the thickness gradient from upper (left) to lower (right) terminal heights. **g** Histogram of branches per afferent for afferents that branched, colored by afferent type. **h** Left: Estimated proportions of vibrissal innervation composition (unmyelinated axons are lower bound estimate). Right: Afferent type proportions of the myelinated innervation subpopulation ($n$ = 149 classified axons).

afferent shows the distinct thin ending feet of Merkel afferents, the entering of Merkel afferents through the glassy membrane and, the characteristic termination pattern of a lanceolate endings outside the root sheath on the glassy membrane (Fig. 3b right and Fig. S3). As the filigree connections of Merkel axons to the ending feet were not resolvable for all Merkel afferents, we chose to only show the root sheath entering of afferents in Fig. 3b (left, white arrow). Club-like endings ($n$ = 52) are the only axonal population terminating in direct apposition to the ringwulst on the glassy membrane in flat terminals (Fig. 3c, left). Termination patterns of club-like endings are restricted to the upper ringwulst edge, likely resulting in pinching of the afferent terminals upon shaft deflection (Fig. 3c, arrows right). Ruffini-like afferents ($n$ = 25) comprise the cavernous sinus level terminal population, target the vibrissal shaft in almost horizontal trajectories and leave the axon arms 100–300 μm below ringwulst height (Fig. 3d). A synopsis of afferent properties enforces the view that we are dealing with few (four) distinct subclasses of myelinated afferents supplied by the deep vibrissal nerve (Fig. 3e). Axon diameter of the four classified afferent types differ significantly between Merkel-club, Merkel-Ruffini, lanceolate-Ruffini and club-Ruffini afferents (Fig. 3f; Kruskal Wallis: $H$ = 38.85, $p$ = 1.8 × 10$^{-8}$; Significance according to Dunn's Bonferroni corrected pairwise comparison with alpha = 0.05), in line with our observation of proximal-distal conduction velocity gradient (Fig. S4). We observed no systematic relation between axon diameter and angular terminal location (Fig. S4), pointing towards high velocity and precise timing for distally terminating afferents. Branching differed systematically between afferent types (Fig. 3g). Of the comprehensively reconstructed vibrissal innervation, 5% correspond to superficially supplied circumferential axons, 32% to deep vibrissal unmyelinated and 63% to deep vibrissal myelinated innervation (Fig. 3h left). Of the putatively complete myelinated axons we estimate Merkel and club-like afferents to be most abundant (35% each), followed by Ruffini-like (17%) and lanceolate endings (13%) (Fig. 3h right). Our rich data set allows distinguishing four sets of afferent types.

### Large caliber and long Ranvier internode distance club like afferent subtypes

While identification of afferent types by morphology is robust and well-established, our three-dimensionally coherent dataset also allowed an analysis of afferent properties unbiased by classic morphological categories. First, we performed a factor analysis of mixed data (FAMD), considering fiber properties (fiber diameter, internode length, vertical terminal height) and mechanically relevant afferents features (termination on/behind the glassy membrane, ringwulst apposition) to reduce dimensionality of our high-dimensional afferents dataset (Fig. 4a). Afferent subtypes aggregated in proximity, yet a clear distinction was not evident. To test for a more fine-grained afferent distinction, we calculated similarity indices (Gower's dissimilarity) of afferents based on axonal parameter (same as in the FAMD) and compared it to analysis based on purely morphological parameter (same parameter as scored in Fig. 3e). While closely resembling each other, axonal parameter indeed revealed similarities of axon groups which escaped our attention through purely morphological identification (Fig. 4b). Next, we performed hierarchical clustering (Wards method) based on afferent similarity indices, which again reproduced our four morphological afferent classes (Fig. 4c). In addition, our clustering revealed sub-cluster of club-like afferents (Fig. 4c). Comparison of the two club like sub-cluster revealed significant differences in fiber diameter and Ranvier internode length (Fig. 4d). Visualization of club like subtypes in the 3D dataset indicates no spatial aggregation (Fig. 4e). Instead, both subtypes evenly tile the ringwulst space (Fig. 4f), and show no prefence for specific axon-arms. We conclude that analysis of axonal properties confirms afferent categories established by classic anatomical work and delineates a previously unrecognized subclass of club-like afferents.

### Preferential innervation dorso-caudal to the shaft

Rats whisk vibrissae back and forth in the plane of the vibrissa row[7,8] (Fig. 5a). We reconstructed the bulk of a C2-vibrissa intrinsic musculature (red in Fig. 5b) from microCT data (Fig. 5b); the contraction of which leads to vibrissa protraction[31–33] (red arrow; Fig. 5b). Vibrissa retraction results from tissue elasticity and a smaller set of so-called extrinsic whisker muscles anchored outside of the whisker pad[32] (not visible in Fig. 5b). Accordingly, the vibrissa musculature and motorneurons[34] are laid out for the controlled protraction of individual vibrissae and protraction-induced contacts[35,36] mediate the minimal impingement strategy[11]. Various studies documented a systematic relationship between afferent activity and whisking behavior[36,37]. We aligned the C2-vibrissa follicle to whisker pad geometry from microCT data to infer protraction direction in the C2-ringwulst and innervation rendering (Fig. 5c). Afferents appear radially polarized to the caudal circumference of the follicle. The analysis of the angular position of deep vibrissal afferent endings (Fig. 5d) confirms the impressions from the whole-mount axon view. We performed a quantitative analysis of afferent polarization along different angles orthogonal to the vibrissal shaft. Interestingly, this analysis revealed that polarization is most significantly different from chance for both myelinated and unmyelinated axons, when comparing angles 180° around the ringwulst aperture to those opposite to it (Fig. 5e). As angular tuning upon deflection and terminal afferent location overlaps for most afferents[21], afferents are positioned to preferentially sample contacts resulting from protraction-induced objects contacts. Sorting of afferent types by terminal angular position reveals even tiling of Merkel and club-like endings and slight polarization of lanceolate and Ruffini-like afferents to the ringwulst aperture, which is naturally devoid of ringwulst

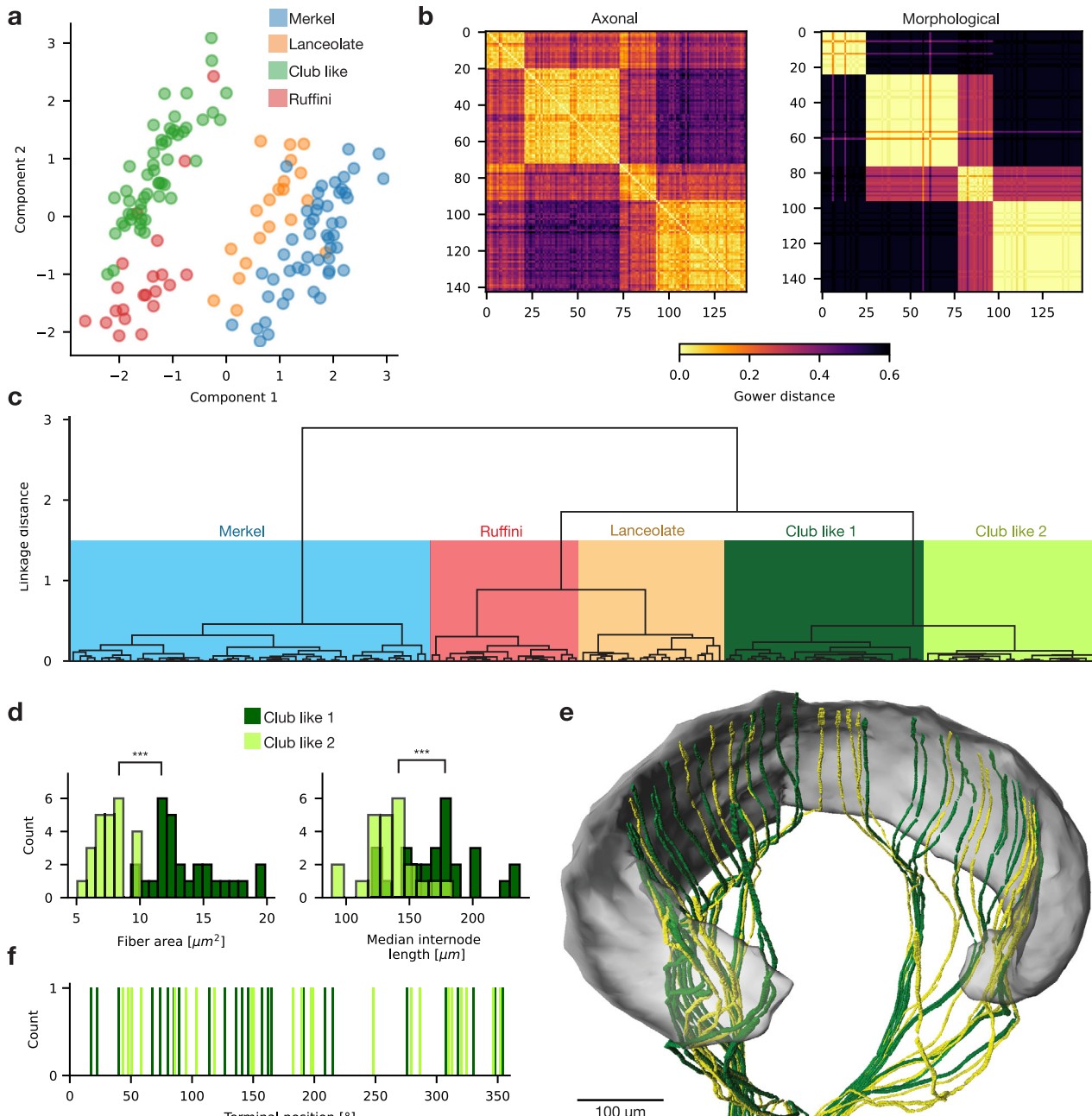

**Fig. 4 | Unsupervised hierarchical clustering supports classical afferent categories and reveals club like afferent subtypes. a** A factor analysis of mixed data of myelinated afferents based on intrinsic parameter colored by morphologically identified afferent type (Merkel = blue, lanceolate = orange, club like = green, Ruffini = red). **b** Left: Gower dissimilarity matrix of afferents based axonal parameter (ringwulst termination, shaft entry, vertical terminal height, fiber diameter, median internode length and branch count). Right: same as left but based on morphological parameter (same as used for morphological scoring in Fig. 3). Both matrices show distinct afferent groups with high similarity, which closely resemble each other. Color bar applies to both plots. **c** Hierarchical clustering (Wards method)

dendrogram from Gower dissimilarity indices based on intrinsic parameter. Biggest cluster match morphologically identified afferent classes (Ruffini, red and lanceolate, orange). Morphologically identified club-like afferent types both separate into two further sub-cluster (club like 1 and club like 2 in dark and light green). **d** Histogram of club like afferents fiber area and Ranvier internode length colored by club like subclass. Asterisk' refer to significance according to two-sided t-test, comparing both distributions (Fiber area: $p = 1.97 \times 10^{-12}$, Median internode length: ip = $2.35 \times 10^{-6}$). **e** 3D rendering of club like subtypes and ringwulst. **f** Histogram of club like subclass positions along the vibrissal circumference colored by club like subclass.

exclusive club-like afferents (Fig. 5f). We conclude that the bulk afferent innervation is polarized to dorso-caudal angles.

**Angularly discrete axon arms and angular follicle linearization in the nerve**

Afferents ascend in conspicuous axon arms, which innervate discrete angular territories, revealed by an arm-based color code (Fig. 6a). While there is little angular order within arms and the constituting

axons often even wrap around each other, axons from neighboring arms only rarely show angular crossings (Fig. 6b). Arms cover roughly similar angular territories (Fig. 6c). Because our large-scale axonal reconstructions reach into the nerve, we could assess how follicle information maps into the nerve. Again, axon arms appear to be decisive units (Fig. 6d). A section through afferents just below the ringwulst reveals their radial arm arrangement around the shaft (Fig. 6e). We noted that afferents ascend/descend orderly and hence

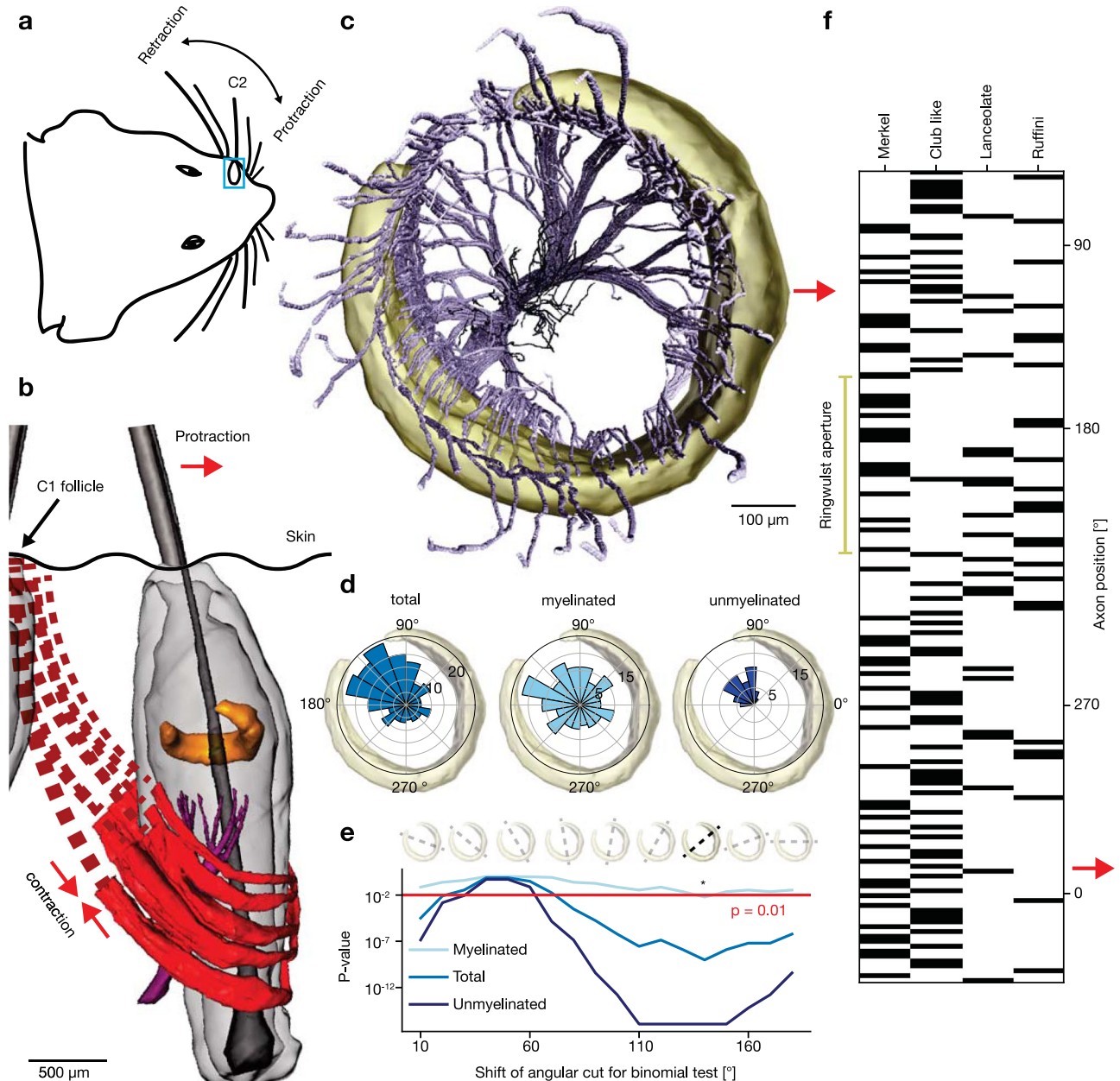

**Fig. 5 | Polarized innervation caudal to the vibrissa. a** Schematic rat head during whisking. Blue square indicates the C2 vibrissa follicle. **b** MicroCT 3D rendering of a C2 follicle and musculature (red) with movement schematic. Contraction of the sling-like muscles induces vibrissa protraction (red arrow). **c** Volume rendering showing asymmetric axonal innervation and ringwulst (top view; red arrow = protraction direction) of the C2 vibrissa. Data of the left C2 vibrissa was mirrored to follow right = forward conventions. $N = 232$ axons (174 myelinated axons in light blue and 58 unmyelinated axons in dark blue). **d** Angular distribution of afferent terminal positions. Innervation is significantly denser caudal (90–270°) than rostral. Myelinated axons contribute weakly (light blue) and unmyelinated axons strongly (dark blue) to the polarization effect. **e** To assess the statistical significance of axonal polarization bias we applied a two-sided binomial test, comparing the afferent distribution of opposing semi-circles for myelinated (light blue), unmyelinated (dark blue) and total innervation (blue). The x-axis represents a cut that separates the semi-circles, which shifts in 20° increments clockwise (as indicated above). Asterisk indicates the axis in which distribution is significantly different from chance for both myelinated and unmyelinated afferents (myelinated: $p = 0.006$, unmyelinated: $p = 1.11 \times 10^{-16}$, total: $p = 9.68 \times 10^{-10}$). **f** Raster plot of afferent types sorted by angular position.

expected a topographic representation of axon arms in the vibrissal nerve (i.e., a radial configuration of arms represented in the nerve) as topography is a landmark feature of sensory systems; this is not, what we found, however. Instead, a section of the representation is introduced (Fig. 6e) and a linear representation of axon arms (which take the shape of bands) is observed in the nerve (Fig. 6f). As a result of this follicle linearization rostral-ventral angles are represented at the distant poles of the nerve. To quantitatively assess linearization of afferents according to terminal position, we tested two idealized models of

afferent terminal layouts in the nerve: (i) 'radially-condensed' model, where the afferent terminal positions are predicted by their radial angular position in the nerve and (ii) a 'linearly-unwrapped' model, where the afferent sequence within the nerve cross-section predicts the terminal position (Fig. 6g). We plotted our predictions against the actual terminal position of the afferents and observe a strong and highly significant correlation for the 'linearly-unwrapped' model (Fig. 6h), while the 'radially-condensed' model did not predict the actual terminal positions well. Axon arms form angular units of follicle

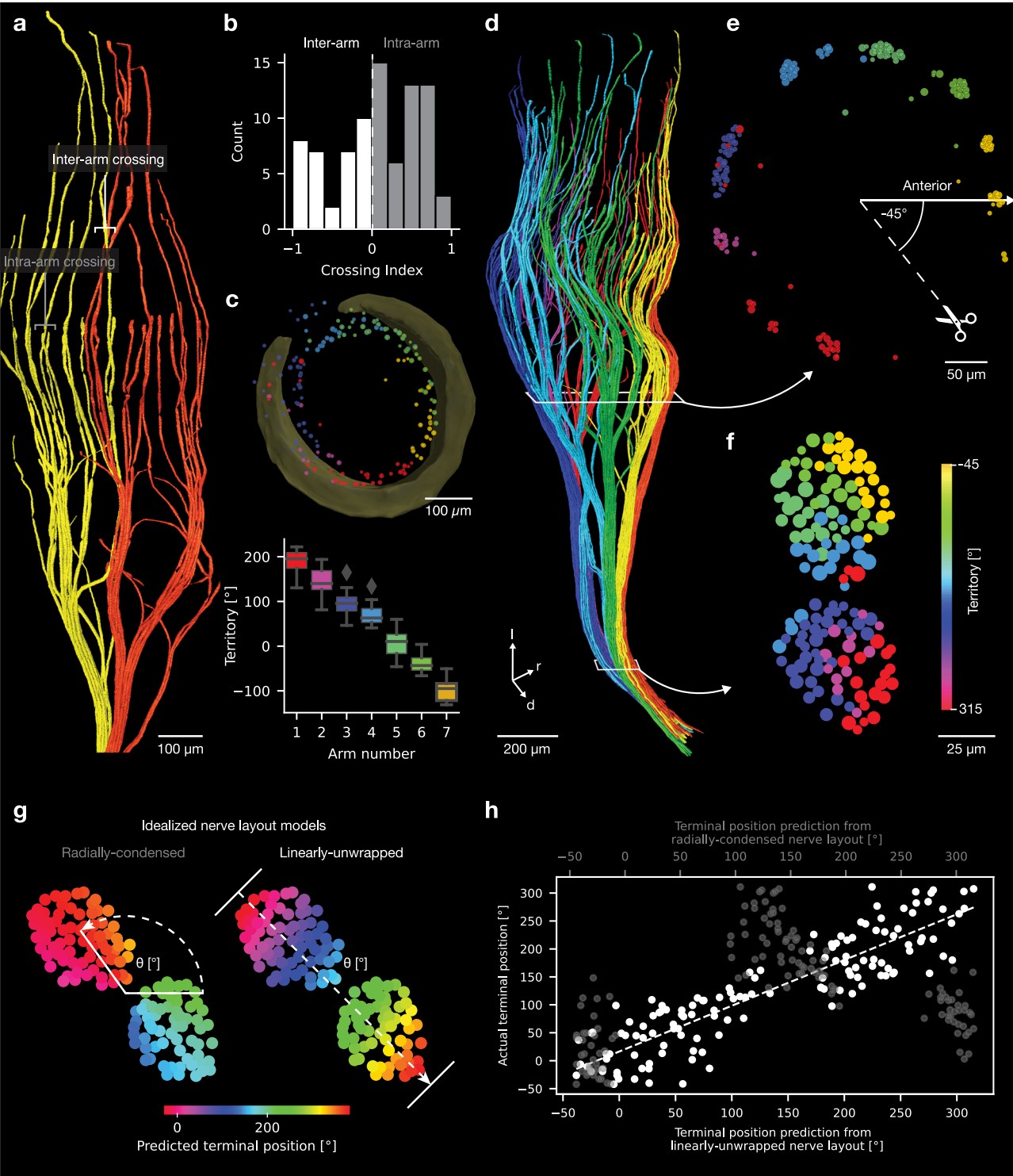

**Fig. 6 | Axon arms as angular units and radial to linear transformation of axon arms in the vibrissal nerve. a** Volume rendering showing two axon arms (yellow and red). Inter-arm axon crossing is rare, while intra-arm axon crossing occurs more frequent. **b** Histogram of axon crossing index distribution. Negative values indicate more inter-arm and positive values more intra-arm crossings (see **a**) per axon. **c** Top, terminal axon positions color coded by axon arm. Bottom, angle territories of axon arms. Boxplots display the 25th to 75th percentile range as the box and the median as center line. Boxplot whiskers extend by the inter quartile range. Outliers are plotted individually. **d** 3D rendering of axonal innervation (color coded by axon arm). White boxes indicate sections shown in (**e, f**). **e** Follicle arm section (see **d**), showing radially arranged axon arms and the cut of the radial representation

introduced in the nerve (see **e**). **f** Nerve section (see **d**). Note the linearization of the radial axon-arm representation seen in (**e**). Axon diameter in (**e, f**) are proportional but not on scale. l lateral, r rostral, d dorsal. **g** Idealized models of afferent terminal angular position layouts in the vibrissal nerve. Left: Terminal angular positions are represented as a condensed angular (radial) arrangement in the nerve cross section. Right: Terminal angular positions are represented linearly along the nerve cross section. **h** Predicted afferent terminal angular position by the radially-condensed model (grey) and the linearly-unwrapped model (black). The y-axis corresponds to the actual terminal afferent position. R- and p-value refers to linear regression for the linearly unwrapped model (black).

innervation and arms are mapped into an orderly linearized representation in the nerve.

## Discussion

X-ray phase contrast tomography data allowed a comprehensive reconstruction of myelinated axons innervating the rat C2-vibrissa follicle. A limitation is that unmyelinated axons could not be fully reconstructed. We observe a relation of axonal diameter and myelination with proximal-distal position of the respective afferent ending. We classified axons by vertical ending territory and terminal morphology into Merkel, lanceolate, club-like and Ruffini-like afferents, building upon previous work on myelinated axons[21–23,30]. Further, higher resolution scans might be required to assess how our descriptions of unmyelinated axons and previous work on unmyelinated axons are related[38].

We find that afferents ascend vertically in ordered straight trajectories with little proximal branching. This architecture results in a high degree of axonal angular specificity and presumably contributes to afferent direction selectivity. We expect that an improved characterization of the vibrissa follicle biomechanics will lead to an improved understanding of coupling and response properties[39].

Vibrissal afferents can respond with astounding temporal precision[15,16] and we find a systematic relation of afferent diameter and vertical termination. We predict very fast and temporally precise transduction events to occur in distally innervating Merkel and lanceolate endings.

Viewing afferent architecture in the context to mechano-accessorial follicle structures suggests that different activation mechanisms apply to each afferent type. The high branching number of lanceolate endings might reduce their angular specificity as observed in identified afferent recording[21]. The comparably smaller angular scattering and even angular tiling of Merkel afferents points towards high angular specificity, which aligns with previous work[21]. In line with earlier observations by Tonomura et al.[30], we observe precise terminal alignment and apposition to the ringwulst of club like afferents; with all likelihood club like afferent activity communicates ringwulst mechanics. Finally, it appears to us that the activation mechanics of Ruffini afferents is quite distinct from all other afferents, as Ruffini afferents approach the shaft in an almost horizontal trajectory.

We observe two classes of club like endings through unsupervised clustering of afferents based on axonal parameters. Classes differ in fiber area and Ranvier node spacing. Club like endings occur exclusively at the ringwulst, which is lacking in non-vibrissa hair follicles and in vibrissa follicles of marsupials[40], hence this afferent type likely evolved after lanceolate and Merkel afferent types. Given that termination patterns of club like endings on the glassy membrane and presence of a terminal Schwann cell[29,30] remind of lanceolate endings, we speculate that the club like endings evolved from the lanceolate afferent type. Specifically, we suggest that the large caliber and long internode subtype occurred earlier in evolution as it more closely resembles axonal parameter of (large caliber) lanceolate afferents.

Our dense axonal reconstruction revealed a significantly denser innervation dorso-caudally to the hair shaft. Since many afferents respond to compression by the whisker shaft[21], this arrangement leads to a preferential sampling of protraction-induced object contacts. Polarized innervation might be a neural adaptation to whisking and we predict less polarized innervation patterns in vibrissae of non-whisking species. Dorso-caudal innervation bias aligns with data from trigeminal ganglion recordings, which revealed a predominance of responses to backward deflections[41] and brainstem, recordings which revealed a predominance of responses to upward deflections[42]. The even angular distribution of Merkel and non-Merkel afferents reminds us of the systematic tiling (coverage) of retinal space by retinal neurons[43]. In contrast, the angularly polarized innervation of lanceolate afferents resembles the polarized innervation of the hair follicle[44].

Four observations suggest axon arms form functional units of follicle innervation: (i) afferents form discrete axonal bundles at the midlevel of the follicle, (ii) afferents within an axon-arm mingle, (iii) afferents of different axon arms innervate discrete angular territories and angular separation is maintained in their terminal fields, (iv) axon-arm order is maintained in the form of discrete bands in the vibrissal nerve. We are surprised that follicle organization in axon arms has received little attention so far.

We observed an elaborate microtopography of the vibrissal nerve. It appears the nerve functions more like an afferent map than a loose cable bundle. Vibrissal nerve microtopography aligns with the presence of a macrotopography in the trigeminal nerve[45,46] and ganglion[41]. We also found a linearization of radial follicle topography in the vibrissal nerve. The elegant radial to linear transformation of afferent topography was entirely unexpected to us. The result shows how large-scale reconstructions of afferent populations can elucidate the neural mapping of sensory information. We suspect that follicle linearization in the nerve might be fundamental to vibrissal representations downstream from the follicle. Specifically, we suggest that the linearized representation of afferents in the nerve informs the linear (tube-like) downstream brainstem barrelettes[47,48]. We are unaware of alternative explanations for the linear, elongated architecture of brainstem barrelettes, which is quite unexpected since vibrissa follicles are often conceptualized as point-like sensory information sources and since sensory representations tend to be compact. Our suspicion that barrelettes mirror the linear angular representation in the nerve is strengthened by the fact that thalamic vibrissa representations (barreloids)[49] also contain an anatomically linearized representation of angular tuning[50] matching the one observed here in the vibrissal nerve. If correct, our linearized-follicle to linear barrelette hypothesis predicts that we find forward-downward direction response preferences at the rostral and caudal poles of barrelettes and backward-upward response preferences at barrelette centers.

We conclude that large-scale reconstructions provide an avenue for understanding vibrissal mechanotransduction and mapping of vibrissal information onto the brain.

## Methods

### Animal specimen

All animal specimens have their origin in animal waste from other experiments. The experiments from which we received cadavers were carried out according to German law for animal welfare and approved by the State Office for Health and Social Affairs committee (LAGeSo) in Berlin (Animal license number: G0279/18, G0095/21) and were killed according to the specific animal experimentation permits. In brief, animals received an initial anesthesia via isoflurane until termination of reflexes, followed by an urethane injection (6 g/kg body weight). Subsequently, animals were transcardially perfused with 4 °C 0.1 M phosphate buffer saline, followed by 4% paraformaldehyde (PFA). Heads were removed from the body and stored in 4% PFA at 4 °C until further use.

### Follicle extraction

The C2 follicles were extracted from the heads of male 7-week-old Long-Evans rat cadavers. First, the whisker pad including the musculature was generously removed via scalpel incisions following the border of the whisker pad. Next, the whisker pad was placed with the lateral side facing downwards and the C2 vibrissa follicles were located. Using fine tweezers, musculature and connective tissue surrounding the C2 follicle were carefully removed. The isolated follicle was then removed from the pad by small incisions at the skin.

### Iodine staining

Whole-whisker pads including skin, musculature and all macro-vibrissae and some micro-vibrissae were stained in 1% Lugol's solution (Morphisto, Cat. #10255) buffered in 0.1 M phosphate buffer (PB)

for 96 h at room temperature on a shaker. To remove access staining solution, samples were washed in 0.1 M PB overnight preceding the microCT scans.

## MicroCT imaging

Whole-whisker pads were haltered in 15 ml falcon tubes and stabilized with 4% agarose. MicroCT scans were acquired with a standardized YXLON FF20 CT system (YXLON International GmbH, Hamburg Germany) equipped with a Perkin Elmer Y Panel 4343 CT detector and 190 kV nano focus transmission tube. One thousand eight hundred transmission images were step-wise obtained over 360° rotation, with 1 s exposure at 60 kV and 90 μA. Tomographic reconstruction was performed using the built-in YXLON reconstruction software with standardized settings.

## Follicle staining

The extracted C2 vibrissa follicle underlying the complete dataset was stained in 1% $OsO_4$, 0.1 M PB overnight at 4 °C. Subsequently, the sample was washed in 0.1 M PB for 24 h to remove excessive staining. The follicle underlying our 50% reconstructed dataset was stained using an en-bloc electron microscopy staining protocol[51]. In short: the extraceted follicle was stained sequentially (i) 2% $OsO_4$, 0.15 M sodium cacodylate buffer (Cac) for 21 h, (ii) washed in 0.15 M Cac for 6 h, (iii) 2.5 % ferrocyanide in 0.15 M Cac for 15 h at 4 °C, (iv) 0.15 M Cac, (v) 2% $OsO_4$ in 0.15 M Cac for 3 h, (vi) 0.15 M Cac for 1.5 h, (vii) $H_2O$ for 2 h, (viii) 4% pyrogallol in $H_2O$ for 5 h, (ix) $H_2O$ for 3 h, (x) 3% uranyl acetate overnight at 4 °C and (xi) 3% uranyl acetate for 1 h at 45 °C. All other steps were performed at room temperature.

## Paraffin embedding

Stained and unstained follicle samples were dehydrated in an ascending ethanol series of 25/50/70% Ethanol in $H_2O$ for 2 h per step and left in 70% ethanol overnight at room temperature. Subsequently, samples were placed in tissue cassettes and placed in Hypercenter XP Tissue Processing system for paraffin infiltration. The program set for paraffinization runs through an ascending ethanol (80/92/96/96/96/96/100) series, a 100% isopropanol and three 100% Rotinistol all at 39 °C and finally three 100% paraffin steps at 60 °C. All steps run under applied vacuum for homogenous infiltration. Fully infiltrated paraffin samples were then placed in 200 μm pipette tips and carefully filled with liquid paraffin on a Leica EG1160 tissue embedding station.

## X-ray phase contrast tomography

The stained incomplete follicle volume and the unstained follicle volume were positioned vertically in 200 μm pipette tips filled with paraffin. The complete and osmium-stained vibrissa follicle was extracted from a paraffin block with a 1.5 mm biopsy puncher and inserted into a 1.5 mm polyimide tube. Volumetric data was acquired by propagation-based X-ray phase-contrast tomography with an unfocused, quasi-parallel synchrotron beam (PB) at the GINIX end station with a photon energy Eph of 13.8 keV, selected by a Si(111) monochromator. Projections were recorded using a microscope detection system (Optique Peter, France) with a 50-m-thick LuAG: Ce scintillator and a 10× magnifying microscope objective onto a sCMOS sensor (pco.edge 5.5, PCO, Germany). This configuration enables a field of view (FOV) of 1.6 mm × 1.4 mm for each projection, sampled at a pixel size of 650 nm. The continuous scan mode of the setup allows the acquisition of a tomographic recording with 3000 projections over 360° in less than 2 min. After recording of the tomographic dataset, dark field and flat field images were acquired. For each follicle, three topographies with an overlap of ~20% were acquired.

## Phase retrieval and tomographic reconstruction

First, the raw detector images were corrected by dark subtraction and empty beam division. Phase retrieval was performed for each projection, using the linear CTF approach[52,53], implemented in the HoloTomoToolbox[54]. Apart from phase retrieval, the HoloTomoToolbox provides auxiliary functions, which help to refine the Fresnel number or to identify the tilt and shift of the axis of rotation[54]. Tomographic reconstruction of the datasets was performed by the ASTRA toolbox[55,56], using the iradon function and a Ram-Lak filter.

## Image post-processing

The whole C2 vibrissa follicle extended over three tomographies, which were stitched using the FIJI pair wise stitching plugin[57]. After importing the whole C2 follicle dataset into Amira, we cropped it to the outer follicle boundaries, which resulted in the depicted volume image in Fig. 1.

## Segmentation

Tomographic images were segmented in an extended version of the Amira software (AmiraZIBEdition 2022.17, Zuse Institute Berlin, Germany). A combination of the 'lasso' and 'brush' tools was used to manually label axons and follicle structures. Labels were placed every 5–50 images and interpolated in between. Manual traceability of axon trajectories largely relies on 1. The quality of the data set and staining, i.e., the osmium-stained data sets that we focus on in our article is much easier (and more reproducibly) reconstructed than the unstained other data set. 2. Observer experience plays a great role in our reconstructions. This is the case, because our axonal tracings do not rely on antibody staining, but merely on axonal contrast and continuity in the 3D data set. 3. Observer identity (who traces) appears to have a minor effect, i.e., tracings from experienced observers in good data sets are largely identical.

## Data analysis and visualization

Data was analyzed and plotted using the matplotlib and seaborn plugins for python 3.6 (https://www.python.org). 3D reconstructions were visualized in Amira. Figures layouts were prepared with Adobe Illustrator (version 28.1).

## Angle measurements

Mid-points of the axon terminals were noted as coordinates within the image data and centered, with the vibrissal shaft midpoint at ringwulst level as ground zero coordinate. Next, we computed the slope of a line between the afferent terminal position and the vibrissal shaft center:

$$a = \frac{y_{afferent\ terminal} - y_{shaft\ center}}{x_{afferent\ terminal} - x_{shaft\ center}}$$

From the slope we calculated the afferent terminal angular position by taking the arctan:

$$\theta = \tan^{-1}(a)$$

## Afferents exclusion criteria

For afferents classifications, we excluded all axons that we lost during segmentation within their axon arms. Axons that showed a clear trajectory and left the axon arms, or when in direct proximity to the root sheath, were included in the analysis, since we presumed those axons to be close to their terminal region. We justify our exclusion criteria as follows: (1) Axon trajectories are mostly curved while within axon arms. (2) Axons that leave the axon arms show a clear trajectory toward the shaft, which indicates their terminal territory. (3) As soon as macroscopic arms disappear and axons run individually, their trajectories are mostly straight and angular territories don't change much. Thus, our exclusion criteria reduce our axon data to those axons that are at or close to their terminal regions. Further, we excluded axon fragments, which could not be connected to a 'ground-cable' from our axon-based

analysis, since we could not sufficiently exclude them as mere branches of another axon. Axon paths that start 100 μm below the ringwulst were considered as 'ground-cables', as branching only occurred upwards of this position.

### Afferent type classification

We classified afferent types based on a multidimensional scale, which comprised vertical ending territory (divided into ring sinus-, ringwulst- and cavernous sinus-level territories) and ending morphologies (visible root sheath entrance, glassy membrane adjacency, branching, multi-branching (>2 branches) and trajectories toward the shaft). Based on these criteria afferent types were classified by their dominating characteristics as: Merkel afferents (ring sinus-level terminal, root-sheath entrance), lanceolate afferents (ring sinus-level terminal, on glassy membrane terminal), club-like afferents (ringwulst terminal, on glassy membrane terminal), and Ruffini-like afferents (below ringwulst terminal, visible trajectories toward the root sheath). In our classification raster plot we only included those afferents that are connected to a 'ground-cable'. Based on our reconstruction of 58 unmyelinated axons, we made a lower-bound estimate of 90 unmyelinated axons for the innervation composition pie chart in Fig. 3h.

### Statistical analysis

Axon polarization: Binomial test was performed using the scipy binomial function. Success rate was set for 0.5 (assumption of equal afferents distribution) and n = numbers of total afferents (see exclusion criteria). Kruskal–Wallis test and post-hoc Dunn's pairwise comparison was performed using the scipy kruskal and scikit-posthocs functions. Linear regression for axon area–terminal height dependency was calculated using the scipy linregress function.

### Axon crossing index

As a metric of axon crossing, we formed pairs of all axons and compared their radial position at the z-position 100 μm below the ringwulst and at the respective axon terminal positions. A sign change was counted as a positive crossing. Based on the crossing count per axon with inter-arm-axons and intra-arm-axons we computed a crossing index I:

$$I_{crossing} = \frac{(N_{intra-arm} - N_{inter-arm})}{N_{total}}$$

Hence, a positive crossing index indicates an axon that crosses more frequently within an axon arm.

### Axon conduction velocity estimation

We estimated conduction velocities of myelinated axons as axon diameter multiplied by a factor of 6/s[58]. Conduction velocities of unmyelinated fibers were computed by a logarithmic function, which we fitted onto datapoints of conduction velocities for unmyelinated fibers in Waxman and Bennett[28] using the NumPy Polyfit function for logarithmic fits, resulting in conduction velocity $v$ as follows:

$$v = 0.32 \cdot \log(fiber\ diameter) + 2.8\ s^{-1}$$

### Factor analysis of mixed data (FAMD)

Because our data consists of partially binary and partially continuous data, we performed a FAMD to reduce dimensionality and visualize our high-dimensional dataset. In our analysis, we included parameter that are important to afferent function including ringwulst adjacency, shaft entry (both binary values), terminal height, median fiber diameter, median Ranvier internode length and branch count. FAMD was performed using the Prince Python library.

### Gower dissimilarity index

To assess afferent similarities, we calculated a Gower distance of all afferent pairs, which measures similarities of mixed-type objects. This was again important, because our data consists of partially binary and continuous data. We calculated the Gower distance on the same afferent parameter as the FAMD as a measure for intrinsic afferent similarities (see above). For comparison, we also calculated a Gower distance of afferent characterized by morphological parameter as in Fig. 3e. Gower distance calculation was performed using the Gower Python library.

### Clustering

Based on the calculated Gower distances of afferent pairs on intrinsic parameter, we performed standard hierarchical clustering (Ward's method) to reveal afferents clusters. Hierarchical clustering was performed using the scipy.hierarchical Python library. We inspected the resulting cluster visually and analyzed the distribution of median axon area and median Ranvier internode lengths of the two large Merkel and club-like subcluster.

### Linearization analysis

To assess the nerve layout, which best predicts the actual terminal angular positions, we compared two nerve layout models. We considered (i) an 'unwrapped-linear' (the radial distribution of afferent terminals is cut at one position and angles are spread out along a linear line) and (ii) a 'condensed-radial' (angular positions of the afferent terminals correlate with the angular position of the respective afferents in the nerve) model, we first performed a linear regression on the afferent nerve positions in the nerve. Next, we projected the afferent positions onto the linear regression line and plotted these x-projections against the actual terminal angular position of the afferents. A linear regression on this data resulted in high correlation ($r = 0.9$, $p = 5 \times 10^{-49}$), supporting the 'unwrapped-linear' model. Next, we calculated angular positions of afferents in the nerve and plotted these angles against the actual terminal positions. This resulted in a very weak correlation value and bare significance ($r = 0.27$, $p = 0.001$).

### Reporting summary

Further information on research design is available in the Nature Portfolio Reporting Summary linked to this article.

## Data availability

We provide all data and figures under the g-node repository https://gin.g-node.org/elephant/Follicle_Innervation. Compressed version of the original source images will be uploaded upon publication. Source data are provided with this paper.

## Code availability

No code central to the conclusions was developed.

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

## Acknowledgements

We thank Matias Mugnaini and Andreea Neukirchner for comments on the manuscript and Undine Schneeweiß, Markus Osterhoff, Peer Martin, Miguel Concha-Miranda and Lennart Eigen for technical assistance. B.G. and M.B. are supported by BCCN Berlin, Humboldt-Universität zu Berlin and the Deutsche Forschungsgemeinschaft (DFG, German Research Foundation) under Germany´s Excellence Strategy – EXC-2049 – 390688087. T.S. and J.R. receive funding by SFB1456/A3 Mathematics of Experiment. We acknowledge DESY (Hamburg, Germany), a member of the Helmholtz Association HGF, for the provision of experimental facilities. Parts of this research were carried out at PETRA III and we would like to thank Fabian Westermeyer for assistance in using the P10 GINIX endstation. Beamtime was allocated for proposal I-20220980. B.G. is supported by Wübben Foundation Wissenschaft.

## Author contributions

Conceptualization: B.G. and M.B.; methodology & materials: B.G., J.A., J.R., T.S., and M.B.; investigation: B.G. and M.B.; formal analysis, B.G. and M.B.; writing: B.G., J.R., T.S., and M.B.; supervision: T. S. and M.B.; funding acquisition: T.S. and M.B.

## Funding

## Competing interests

The authors declare no competing interests.
