## [Transparent Peer Review file · Nature Communications]

Three-dimensional architecture and linearized mapping of vibrissa follicle afferents

Corresponding Author: Dr Michael Brecht

Version 0:

Reviewer comments:

Reviewer #1

(Remarks to the Author)

Beautiful anatomical study of an intriguing sensory organ – the rat's whisker-follicle. The authors reconstruct in detail the 3D morphology and connectivity of afferent sensory receptors within a rat's C2 whisker-follicle. For this purpose, they utilize the excellent spatial resolution of synchrotron x-ray phase tomography. Most results are from one sample, which was osmium-stained. A second, unstained follicle was used for confirmation. The paper describes in an illuminating manner the innervation composition of the four major receptor types, their spatial organization within the follicle and the spatial organization of their ascending axons. The authors found that the height of axon termination correlates with axonal diameter, and by extension – with conduction velocity. When myelinated and unmyelinated afferents are aggregated, afferents are more likely to terminate at an angle caudal to the whisker, i.e. at areas stimulated by whisker deflection from the front of the snout. Axons are arranged in bundles which terminate in the follicle in distinct polar locations, but as they are pooled in the vibrissal nerve, this polar map is translated to a linear one.

The methodological approach is impressive in its ability to reveal innervation patterns of the intact vibrissa follicle-sinus complex in unprecedented detail. The detailed 3D micro-anatomical account of the follicle, presented in the current work, is indeed impossible to obtain using traditional methods of histology.

This excellent work is timely and important. Perception depends crucially on the sensory organs, which are highly complex evolutionary developments. It is commonly agreed that without understanding sensory transduction there is no real chance to understand perception. Nevertheless, our knowledge of the processes through which brains interact with their environments, which are bottlenecked within sensory organs, is very limited. This study takes a significant step in the required direction, by shedding new light on the morphology of the whisker-follicle, from which mechanotransduction can be better understood.

I have several suggestions to improve this already-excellent report, and a list of additional comments.

Major:

1. Circular analysis. The identification of ending type relies, among other factors, on the location of the ending (based on previous knowledge). A significant component of the new results here is the distribution of ending types along the follicle. Not surprisingly, thus, the results show a clear division between the innervation location of the four ending types. It seems that this circular analysis hampers the great advantage of the fantastic new tool presented here. The new tool could be used to validate, in more objective manner, the spatial distribution of the different ending types. For this, the circularity of the analysis should be broken. That is, the authors should conduct an analysis that identifies the ending type without relying on its location, and see if it confirms the results of the circular analysis. In doing so, it is recommended to also use an independent human observer, in a double-blind method, to preclude a-priori biases. Such a confirmation step need not necessarily be comprehensive. Yet, it should be convincing.

2. Functional relevance. A great contribution of this study could be the shedding of new light over structure-function relationships in the follicle. There are a handful of functional types of follicle primary afferents and a handful of ending types, but the relations between them are not known beyond the classical SA/RA and RF-size classification of passive responses. The resolution of the current method, and its superb 3D reconstruction ability, allows for much more than this. Panels in Fig. 3 and in Fig. S2 indicate that the resolution allows for characterizing major differences between the innervation morphology of the different types, enough to suggest potential coding differences. The authors indeed show a slight bias towards caudal innervation, which is consistent with active touch being typically made during protraction. It is suggested that the authors attempt to go deeper into potential differences in mechanotransduction, based on morphological differences and known follicle dynamics during active touch.

Additional comments:

1. An interesting and important piece of information that is missing is the number of endings per afferent fiber (examples, statistics) – It would be an important addition to the current publication.
2. Reconstruction details are slightly unclear and it is hard to say how observer-dependent they are. Please clarify where needed.
3. Please add a ruler along the follicle height, in one of the panels, such that the heights indicated in Fig. 2f-g can be affiliated with the follicle organs.
4. Fig. 2f-g are interpreted as reflecting a monotonic positive dependency of area/velocity on terminal height among the myelinated axons. But another appealing interpretation is a division of myelinated axons to two categories – those terminating below and above ~850 microns. Please provide both alternatives and explain your preferred interpretation if any.
5. Fig. 3f, would be interesting to consider how well ending types could be reconstructed from a combination of features, e.g. axonal diameter and terminal height, once the circularity mentioned above is broken.
6. In the methods “Long-Evans rat” is spelled as “long evan rat”
7. Although findings are often visually very clear, several places are lacking direct quantification of verbally-reported findings. Examples:
 - Section “Follicle axon classes and a diameter gradient of deep vibrissal afferents” has a list of items in Roman numerals, discussing some characteristics of the afferents visualized in fig. 1. For example, that afferents have mostly vertical trajectories. This is clearly true, but it would still be nice if a quantification was made and submitted for significance analysis.
 - Fig. 4d different polarization between myelinated and unmyelinated axons + fig 4e different subgroups by endings: polarization is discussed and interpreted in the text, but not quantified and submitted to significance analysis.
 - Fitting a polar curve to 5e and a linear (or piecewise linear) to 5f would help give quantitative evidence to the interpretation in the text.
8. Fig. 3e – please explain what is shown
9. Authors should define what they mean by the term “axon arm” and use it in only one sense throughout.
10. Listing the main findings of the afferents’ reconstruction, the authors mentioned in the first place “afferents ascend in ordered, straight trajectories” (page 6, line 2). This finding was already described in the mouse follicle as “rows of parallel, ascending bundles of medullated fibers which travel upward” (Melaragno & Montagna, 1953), and this study is not mentioned by the authors.
11. The terms “anterior” and “posterior” are used in the MS to designate coordinates and axes in many places in the text and figure captions. Referring to the coordinates of the rodent mystacial pad, it should be replaced by the terms “rostral” and “caudal”, as it is generally accepted for the animals with linear nervous systems.
12. Fig. 4b shows “bulk of a C2-vibrissa intrinsic musculature” (page 7, 8th line from the bottom). It is known that each follicle is attached to two intrinsic muscles, but here only one muscle is shown, and its insertion is tracked into the “Skin”, which is not accurate because most of the fibers of this muscle are attached to the distal end of the C1 follicle.
13. Caption of Fig. 5d: in “l = lateral, a = anterior, d = dorsal”, replace “a = anterior” with “r = rostral”. There is no need to repeat “l = lateral, a = anterior, d = dorsal” at the end of entire caption.
14. Fig. 5d: replace “a” with “r”.
15. Fig. S3a. Right: In the caption, it is written “ $R^2 = 0.25$, $p = 2.72 * 10^{-10}$ ”, while in the respective place of the figure “ $p = 1$, $19 * 10^{-10}$, $r^2 = 0.25$ ”. It looks reasonable to leave the correct one, and only in the caption.
16. Fig. 5a, red is misspelled as ‘rad’.

Reviewer #2

(Remarks to the Author)

Gerhardt et al seek to better understand the mapping of primary afferents onto the whisker follicle, a structure previously studied in detail at the light-level from Rice and colleagues (Ref. 22) and Kumamoto and colleagues (Structure-function correlations of rat trigeminal primary neurons ..., Proc Japan Acad, Series B 2015). Gerhardt et al apply high resolution imaging with synchrotron X-ray phase contrast computed tomography (CT) to achieve volumetric imaging of the follicle at resolutions previously undescribed for intact samples of an entire follicle. This produced an end-to-end mapping of sensory receptors from their innervation site to the fasciculated axons that leave the follicle.

A strength of the current paper is the use of CT to rapidly generate a 3D dataset of the follicle. Compared to Kumamoto et al (2015), the current effort replaces the equivalent of more than 1000 plastic embedded sections to be collected, stained, and stitched to make a comparable dataset of the entire follicle. Consequently, the path of single axon terminals from the top of the follicle to the bottom across an entire 3D volume has not been performed until this present work.

There are anatomical findings that probably will not change with more samples, but some finding may well change. One is the angular distribution of nerve endings, particularly around the ringwulst (Figs. 3bc, 4cd), which may be essential for understanding the dependence of the trigeminal response to different angles of whisker deflection. For that reason, and potentially other areas of systemic versus unconstrained anatomical variability, a larger sample size, e.g., four or more C2 follicles, must be analyzed to assess animal-to-animal variability**

Interpretive issues:

1. Gerhardt et al should clarify the issue of angular coverage stated in “We conclude that the bulk afferent innervation is

polarized to posterior angles and that Merkel-afferents evenly tile the vibrissa follicle circumference." Our reading is that the Merkel cells are uniformly distributed over angles (Fig. 4e) with complete if nonuniform innervation (Figs. 3b, 4d). If this is correct, why do the authors discuss a preference for contact upon protraction? Is it because the club endings appear to have a bias in angular distribution (Fig. 3c)? Further to the point, if the ringwulst is a constant volume structure, then, gap aside, compressing it on one side will expand it on the other and stretch sensory endings.

2. Gerhardt et al finds a cluster of deep, unmyelinated axons that arrive at the follicle via the deep vibrissal nerve. These neurons preferentially innervate the posterior aspect of the whisker shaft with a tight distribution (Fig. 4d) at the top of the cavernous sinus. These axons are somewhat mysterious to me. Detailed GSA and IHC work of the rat follicle (Comprehensive immunofluorescence and lectin binding analysis of vibrissal follicle ... JCN 1998) showed that the densest innervation site of unmyelinated axons was the inner conical body, distal to this site. However, there also should be unmyelinated axons present along the length of the follicle. These axons are generally described as being widely distributed. The authors need to reconcile this past claim with the somewhat precisely localization of axons in their study. The authors note that "reconstruction of unmyelinated axons was hampered by resolution limitations and is partially incomplete." Given these limitations, contradictions with current literature of unmyelinated axons in the follicle, and one sample, it seems best to exclude this finding of asymmetry from the analysis.

3. Gerhardt et al state that despite the angular distribution of sensory endings at the receptor level of the follicle (Fig. 5e), axons arranged themselves to represent angles linearly rather than radially when they leave the follicle complex (Fig. 5c,f). The authors then speculate this may be the mechanism for achieving linear somatotopy along the log-like structure of the barreloids in thalamus. They also hypothesize that perhaps the unknown receptor directional tuning somatotopy of the trigeminal nuclei barrelettes would be linear by extension. There is no statistical quantification for the claim of linearization. Further, the data is presented with a color-coding that seems to best magnify "linear" somatotopy by grouping single afferents according to their respective "axon arms", or subcollection of neighboring receptors that cluster. Thus while the data in Fig 5c illustrates that there is overlap between the individual receptor locations in each "axon arm", that of Fig. 5f removes this variability. In fact, it appears that the upper fascicles in Fig. 5f codes contact to the caudal part of the follicle, as in contact during protractions, while the lower fascicle codes contact during retraction.

4. Gerhardt et al should comment on how axial forces generated by whisker contact will be detected and .encoded by the different receptor groups.

Minor issues:

1. The synopsis "Afferent responses are tuned to the angular displacement of the shaft, respond with astounding temporal precision providing information about object distance and orientation." should include reference to the awake animal results of Ref. 29, which shows modulation of trigeminal spiking based on mystacial muscle activity and on phase in the whisking cycle.

2. The statement "Vibrissa retraction results from tissue elasticity and a smaller set of so-called extrinsic whisker muscles anchored outside of the whisker pad (not visible in Fig. 4b)." should include Ref. 31.

3. The reference following "... the contraction of which leads to vibrissa protraction" should include Simony et al (J Neurosci 2010), co-authored by Brecht, in addition to Refs. 29 and 30.

4. Please clarify "Characteristic shaft bending moments provide pre-neural information of object touch locations". The bending is a measure of the applied torque, and the associated force will have lateral and axial components. Where is the "pre-neural information" encoded?

5. I appreciate the inability to reconstruct all unmyelinated axons; the IARPA MICrONS EM project has the same issue. Is there a sense that this could affect the conclusion on the anisotropy of innervation (Fig. 4d).

6. The attachment to the intrinsic muscles is shown (Fig. 4b). Yet attachment to the skin and plate (in the notation of Dorfl, Ref. 30) is not determined. This systematic issue limits the mapping of the data to a mechanical model.

Bottom Line:

There are some exciting claims in this work. Unfortunately the reliability of the claims is overshadowed by the lack of replicates and thus the inability to distinguish variability between animals as opposed to purposeful anatomical organization.

**We are aware that Brecht successfully used x-ray CT to look at muscles (Kaufmann et al, Sci Adv 2022; Longren et al, Curr Biol 2023) and innervation (Purkart et al, Curr Biol 2022) of the trunk of the baby elephant in a series of "N = 1" studies. Yet those were pioneering one-off experiments that further relied on the dearth of elephants compared to rats.

Reviewer #3

(Remarks to the Author)

I co-reviewed this manuscript with one of the reviewers who provided the listed reports. This is part of the Nature Communications initiative to facilitate training in peer review and to provide appropriate recognition for Early Career

Researchers who co-review manuscripts.

Version 1:

Reviewer comments:

Reviewer #1

(Remarks to the Author)

Thanks again for sharing with me this wonderful paper – it is pure joy:)

The authors have satisfactorily addressed all my previous concerns and exceeded expectations, producing exciting new results.

Upon re-reading, I have a few additional comments that may help refine and clarify their key messages.

Minor:

Proximal-to-distal gradient in axon diameter. Examination of fig.2f, S2c and S4a suggests to me that instead of a gradient, there are two modes of diameters – small, for < 850 microns, and large for >850 microns. I don't see a gradient in the latter. I suggest that the authors will reconsider the classification here. The difference between a gradient and 2-modes may become important for future evaluations of the evolutionary steps involved in follicle specialization.

Fig. 3e – add in legend that black stands for the presence and white for the lack of a property. In x-axis label write "Axon count" or something alike, rather than "Axon".

Do individual arms contain a mix of the two types of club-like fibers or not? In addition, and in a related manner, could the authors speculate regarding the evolutionary order of the two club-like types?

Reviewer #2

(Remarks to the Author)

I support publication in light of the answers to my queries.

Reviewer #3

(Remarks to the Author)

Three-dimensional architecture and linearized mapping of vibrissa follicle afferents

- Revision round 1 –

Dear editors and referees,

We are most thankful for the fair and constructive criticism of our manuscript. We addressed all concerns raised and the following is a summary of what we have done:

- 1. We added an additional high-quality synchrotron scan of the C2 rat vibrissa follicle. Our main conclusions are all supported now by three samples and our data indicate a low variability of C2 vibrissa innervation architecture.**
- 2. We added a more objective analysis of afferent types by hierarchical clustering of axonal features of afferents. First, clustering analysis confirmed our previous morphological afferent type classifications. Second, this clustering analysis led to the discovery of new club-like afferent subtypes. Hence, this novel analysis provides rigorous quantitative support for the distinction of four deep vibrissal nerve afferent types (Merkel, lanceolate, club like, Ruffini) as suggested by classic anatomical work (Rice and colleagues) and identified afferent recording/labelling (Kumamoto and colleagues).**
- 3. We backed up some of our verbally communicated observations by adding quantification of afferent subtype polarization, branching and axonal trajectories.**
- 4. We provide quantitative evidence for the linearization of vibrissa afferents.**
- 5. We implemented point-by-point all of the referees suggestions as detailed below.**

We think these changes greatly improved our manuscript. We repeat the referee comments in italics prior to responding to them below.

Reviewer #1 (Remarks to the Authors):

Beautiful anatomical study of an intriguing sensory organ - the rat's whisker-follicle. The authors reconstruct in detail the 3D morphology and connectivity of afferent sensory receptors within a rat's C2 whisker-follicle. For this purpose, they utilize the excellent spatial resolution of synchrotron x-ray phase tomography. Most results are from one sample, which was osmium-stained. A second, unstained follicle was used for confirmation. The paper describes in an illuminating manner the innervation composition of the four major receptor types, their spatial

organization within the follicle and the spatial organization of their ascending axons. The authors found that the height of axon termination correlates with axonal diameter, and by extension - with conduction velocity. When myelinated and unmyelinated afferents are aggregated, afferents are more likely to terminate at an angle caudal to the whisker, i.e. at areas stimulated by whisker deflection from the front of the snout. Axons are arranged in bundles which terminate in the follicle in distinct polar locations, but as they are pooled in the vibrissal nerve, this polar map is translated to a linear one.

The methodological approach is impressive in its ability to reveal innervation patterns of the intact vibrissa follicle-sinus complex in unprecedented detail. The detailed 3D micro-anatomical account of the follicle, presented in the current work, is indeed impossible to obtain using traditional methods of histology.

This excellent work is timely and important. Perception depends crucially on the sensory organs, which are highly complex evolutionary developments. It is commonly agreed that without understanding sensory transduction there is no real chance to understand perception. Nevertheless, our knowledge of the processes through which brains interact with their environments, which are bottlenecked within sensory organs, is very limited. This study takes a significant step in the required direction, by shedding new light on the morphology of the whisker-follicle, from which mechanotransduction can be better understood.

I have several suggestions to improve this already-excellent report, and a list of additional comments.

Comment: The referee provides a summary of our findings and comes to an overall very positive assessment of our study. We are very thankful for the appreciation. The referee explicitly acknowledges that our analysis goes beyond the state-of-the art of conventional histological analysis. The referee also notes that he/she has suggestions for improvement.

Change: We address the suggestions for improvement point-by-point below.

Major:

1. *Circular analysis. The identification of ending type relies, among other factors, on the location of the ending (based on previous knowledge). A significant component of the new results here is the distribution of ending types along the follicle. Not surprisingly, thus, the results show a clear division between the innervation location of the four ending types. It seems that this circular analysis hampers the great advantage of the fantastic new tool presented here. The new tool could be used to validate, in more objective manner, the spatial distribution of*

the different ending types. For this, the circularity of the analysis should be broken. That is, the authors should conduct an analysis that identifies the ending type without relying on its location, and see if it confirms the results of the circular analysis. In doing so, it is recommended to also use an independent human observer, in a double-blind method, to preclude a-priori biases. Such a confirmation step need not necessarily be comprehensive. Yet, it should be convincing.

Comment: We thank the referee for this excellent suggestion; as correctly noted, our approach for identifying afferents had a confirmatory bias, because we defined afferents according to previous neuroanatomical work. We also agree that our very rich synchrotron dataset should allow to break this confirmatory or circular bias by an unbiased scoring of afferents.

Change: We implemented an unsupervised analysis of afferent types based on axonal parameters through factor analysis of mixed data, Gower's distance calculation and hierarchical clustering. For our analysis, we chose parameter that are relevant to the mechanical activation of the afferents and to afferent conduction properties, including absolute terminal height, fiber area, Ranvier internode length, branch count, ringwulst adjacency and shaft entry. Indeed, the resulting afferents cluster match with classes obtained through morphological scoring and thus re-confirm the existence of those distinct afferent types from a fiber property point of view (see newly added Figure 4). Further, it revealed subcluster in club afferent classes which are distinct by fiber diameter and Ranvier node spacing. This analysis strengthens our afferents classification and underscores the power of the three-dimensionally coherent view on the entire afferent population. We reckon that this is an important novel result which we present in our newly added figure 4 in the revised manuscript. Besides the discovery of novel subtypes, this analysis provides quantitative evidence for the earlier afferent classification suggested by classical anatomy (Rice and colleagues) and identified afferent recording/labelling (Kumamoto and colleagues).

2. *Functional relevance. A great contribution of this study could be the shedding of new light over structure-function relationships in the follicle. There are a handful of functional types of follicle primary afferents and a handful of ending types, but the relations between them are not known beyond the classical SA/RA and RF-size classification of passive responses. The resolution of the current method, and its superb 3D reconstruction ability, allows for much more than this. Panels in Fig. 3 and in Fig. S2 indicate that the resolution allows for characterizing major differences between the innervation morphology of the different types, enough to suggest potential coding differences. The authors indeed show a slight bias towards*

caudal innervation, which is consistent with active touch being typically made during protraction. It is suggested that the authors attempt to go deeper into potential differences in mechanotransduction, based on morphological differences and known follicle dynamics during active touch.

Comment: The referee suggests a deeper functional analysis of different afferent types and we think this is an excellent suggestion.

Change: We added a paragraph on activation and resulting coding differences of afferent subtypes based on our morphological and fiber property analysis (see line 292 – 302) in the revised manuscript).

Additional comments:

1. *An interesting and important piece of information that is missing is the number of endings per afferent fiber (examples, statistics) - It would be an important addition to the current publication.*

Comment: This suggestion relates to the point above and is an excellent idea.

Change: We plotted number of endings per afferent type and added this as panel g to Figure 3.

2. *Reconstruction details are slightly unclear and it is hard to say how observer-dependent they are. Please clarify where needed.*

Comment: Reconstruction reproducibility depends strongly on: 1. The quality of the data set and staining, i.e. the osmium-stained data sets that we focus on in our article is much easier (and more reproducibly) reconstructed than the unstained other data set. 2. Observer experience plays a great role in our reconstructions. This is the case, because our axonal tracings do not rely on antibody staining, but merely on axonal contrast and continuity in the 3D data set. 3. Observer identity (who traces) appear to have a minor effect, i.e. tracings from experienced tracers in good data sets are largely identical.

Change: We comment on these aspects of data quality in the revised method section on segmentation of the ms.

3. *Please add a ruler along the follicle height, in one of the panels, such that the heights indicated in Fig. 2f-g can be affiliated with the follicle organs.*

Comment & Change: Done.

4. *Fig. 2f-g are interpreted as reflecting a monotonic positive dependency of area/velocity on terminal height among the myelinated axons. But another appealing interpretation is a division of myelinated axons to two categories - those terminating below and above ~850 microns. Please provide both alternatives and explain your preferred interpretation if any.*

Comment: The referee's astute observation of a bimodal fiber area distribution is very interesting. We tested this hypothesis by plotting the distribution of fiber area of <850 μm and >850 μm terminating afferents separately (Referee Figure 2 left). However, this did not result in a clear-cut bimodal distribution by terminal height. Instead, the >850 μm terminating distribution itself appears to contain two distinct peaks. We think the bimodal appearance of the >850 μm terminating fiber distribution rather reflects differences by afferent types (see also Fig. S3). Indeed, when plotting the distribution of fiber area by afferent type (we chose to omit the Ruffini type, because it mostly ends <850 μm), the Merkel and club like types show two peaks in their distribution. This also reflects our findings from unsupervised afferents clustering.

Change: We added these plots to supplementary Figure 2.

5. *Fig. 3f, would be interesting to consider how well ending types could be reconstructed from a combination of features, e.g. axonal diameter and terminal height, once the circularity mentioned above is broken.*

Comment: The referee suggests an interesting analysis, which addresses the point made above relating to the circular analysis in our manuscript. We address this suggestion by our additional unsupervised analysis through hierarchical clustering as we pointed out in our earlier responses above.

Change: See our response above and the additional hierarchical clustering analysis presented in the newly added Figure 4.

6. *In the methods "Long-Evans rat" is spelled as "long evan rat"*

Comment & Change: We fixed this mistake.

7. *Although findings are often visually very clear, several places are lacking direct quantification of verbally-reported findings. Examples:*

- *Section “Follicle axon classes and a diameter gradient of deep vibrissal afferents” has a list of items in Roman numerals, discussing some characteristics of the afferents visualized in fig. 1. For example, that afferents have mostly vertical trajectories. This is clearly true, but it would still be nice if a quantification was made and submitted for significance analysis.*
- *Fig. 4d different polarization between myelinated and unmyelinated axons + fig 4e different subgroups by endings: polarization is discussed and interpreted in the text, but not quantified and submitted to significance analysis.*
- *Fitting a polar curve to 5e and a linear (or piecewise linear) to 5f would help give quantitative evidence to the interpretation in the text.*

Comment: 1. The referee requests a more quantitative treatment of afferent trajectories and we think this is an excellent idea.

2. The referee further notes that our treatment of axonal polarization around the vibrissal shaft lacked quantitative rigor and we agree with this sentiment.

3. The referee also suggests fitting a polar curve to Figure 5e to support our claims. We agree that further quantification should have been added here. We approached this problem differently, however. Please see point 3 of the changes described below.

Change: 1. We quantified to what extent axons ascend the follicle vertically and how this aligns with the vibrissa shaft. We present our findings in our novel supplementary Figure 2.

2. We performed a quantitative analysis of afferent polarization along different angles orthogonal to the vibrissal shaft. Interestingly, this analysis revealed that polarization is most significantly different from chance for both myelinated and unmyelinated axons, when comparing angles 180° around the ringwulst aperture to those opposite to it.

3. We added novel analysis on our observation of follicle linearization by predicting afferent terminal angular positions from (i) the radial angular position of an afferent in the nerve and (ii) the position of the afferents along a linear regression of the afferent positions along the nerve. We find that predicted terminal positions from a linear model aligns well with the actual terminal position, while there is no correlation for predicted terminal positions from a radially laid out nerve.

8. *Fig. 3e - please explain what is shown*

Comment & Change: We added the requested explanation.

9. *Authors should define what they mean by the term “axon arm” and use it in only one sense throughout.*

Comment: The referee suggests defining the term ‘axon-arm’ and to use it more consistently. We agree with the referee that a more stringent definition of axon-arms would be desirable. It is the case that axon-arms are visually obvious, in particular at vertical levels below the ringwulst, but a rigorous definition of axon-arms is complicated and by the lack of a strict consistency of axon-path bifurcations.

Change: We now note in the manuscript that we use the term axon-arm in a cursory fashion to describe the bundling of axons and do not have a stringent axon-arm definition (see line 129 - 134).

10. *Listing the main findings of the afferents’ reconstruction, the authors mentioned in the first place “afferents ascend in ordered, straight trajectories” (page 6, line 2). This finding was already described in the mouse follicle as “rows of parallel, ascending bundles of medullated fibers which travel upward” (Melaragno & Montagna, 1953), and this study is not mentioned by the authors.*

Comment: We overlooked this earlier result and thank the referee for pointing this out.

Change: We now acknowledge the Melaragno & Montagna, 1953 finding in the revised ms (see ref 27).

11. *The terms “anterior” and “posterior” are used in the MS to designate coordinates and axes in many places in the text and figure captions. Referring to the coordinates of the rodent mystacial pad, it should be replaced by the terms “rostral” and “caudal”, as it is generally accepted for the animals with linear nervous systems.*

Comment: We agree.

Change: We adjusted the terminology accordingly.

12. *Fig. 4b shows “bulk of a C2-vibrissa intrinsic musculature” (page 7, 8th line from the bottom). It is known that each follicle is attached to two intrinsic muscles, but here only one*

muscle is shown, and its insertion is tracked into the “Skin”, which is not accurate because most of the fibers of this muscle are attached to the distal end of the C1 follicle.

Comment: The comment of the referee is justified.

Change: We improved our microCT reconstruction and added the missing detail to Figure 5.

13. Caption of Fig. 5d: in “l = lateral, a = anterior, d = dorsal”, replace “a = anterior” with “r = rostral”. There is no need to repeat “l = lateral, a = anterior, d = dorsal” at the end of entire caption.

Comment & Change: We fixed this mistake.

14. Fig. 5d: replace “a” with “r”.

Comment & Change: We fixed this mistake.

*15. Fig. S3a. Right: In the caption, it is written “ $R^2 = 0.25$, $p = 2.72 * 10^{-10}$ ”, while in the respective place of the figure “ $p = 1, 19 * 10^{-10}$, $r^2 = 0.25$ ”. It looks reasonable to leave the correct one, and only in the caption.*

Comment & Change: We fixed this mistake.

16. Fig. 5a, red is misspelled as ‘rad’.

Comment & Change: We fixed this mistake.

Reviewer #2 (Remarks to the Author):

Gerhardt et al seek to better understand the mapping of primary afferents onto the whisker follicle, a structure previously studied in detail at the light-level from Rice and colleagues (Ref. 22) and Kumamoto and colleagues (Structure-function correlations of rat trigeminal primary neurons ..., Proc Japan Acad, Series B 2015). Gerhardt et al apply high resolution imaging with synchrotron X-ray phase contrast computed tomography (CT) to achieve volumetric imaging of the follicle at resolutions previously undescribed for intact samples of an entire follicle. This produced an end-to-end mapping of sensory receptors from their innervation site to the fasciculated axons that leave the follicle.

A strength of the current paper is the use of CT to rapidly generate a 3D dataset of the follicle. Compared to Kumamoto et al (2015), the current effort replaces the equivalent of more than 1000 plastic embedded sections to be collected, stained, and stitched to make a comparable dataset of the entire follicle. Consequently, the path of single axon terminals from the top of the follicle to the bottom across an entire 3D volume has not been performed until this present work.

Comment: The referee provides a summary of our work and correctly characterizes the strength of our analysis, which lies in the dense and complete reconstruction of afferents throughout the whole organ. As correctly pointed out by the referee, our manuscript builds heavily on the pioneering work by Rice et al. (ref 22 in the original and ref 23 in the revised manuscript) and Tonomura, Kumamoto et al. (ref 28 in the original and ref 30 in the revised manuscript), which laid the foundation for our work. Specifically, Tonomura, Kumamoto et al. not only reconstructed club like endings and the ringwulst, but also described their physiological properties; an absolutely ground-breaking contribution to our understanding of vibrissa afferents. This work heavily focussed on club like endings and our work essentially confirmed key findings of Tonomura, Kumamoto et al., namely that club like endings do not branch and that there are on the order of 50 such endings in a large follicle. Our study, in contrast, focussed on the comprehensive reconstruction of all follicle afferents. Our conclusions mainly relate to the overall organization and comparison of different afferents (axon diameter gradient along the follicle, afferent polarization, afferent linearization, derivation of afferent types from unsupervised clustering of afferent properties). Hence, we consider our work and the work of Tonomura, Kumamoto et al. as largely complementary.

Change: None.

*There are anatomical findings that probably will not change with more samples, but some finding may well change. One is the angular distribution of nerve endings, particularly around the ringwulst (Figs. 3bc, 4cd), which may be essential for understanding the dependence of the trigeminal response to different angles of whisker deflection. For that reason, and potentially other areas of systemic versus unconstrained anatomical variability, a larger sample size, e.g., four or more C2 follicles, must be analyzed to assess animal-to-animal variability***

Comment: The referee asks about follicle-to-follicle variability (across animals). We think this is a valid question and hence added a novel C2 vibrissa follicle data set. We also note that our dense analysis of all myelinated follicle afferents is a first-of effort. Our ambition was to provide a first draft of detailed C2 architecture rather than to describe the variability of C2 vibrissa innervation architecture across animals. We think this goal has been fully achieved.

Change: To address the referee's concern of animal-to-animal variability, we added another C2 vibrissa follicle dataset (Supplementary Figure 2), which was stained with osmium and uranyl acetate. Further, we performed a deep quantitative analysis on all three datasets (Supplementary Table 1). We made the following observations:

1. The three datasets have a stunningly similar visual appearance (Figure S1).
2. The number of myelinated axons in all three follicles is very similar (Supplementary Table 1).
3. The proportions of afferent types is very similar across C2 follicles (Supplementary Table 1).
4. The polarization of afferents similar across C2 vibrissa follicles and points toward the ringwulst aperture (Supplementary Table 1).
5. Similar vertical axon diameter gradient is observed in both datasets from stained C2 vibrissa follicles (Supplementary Table 1).
6. In all three C2 vibrissa follicles terminal angular positions sort into a linear arrangement in the nerve (Supplementary Table 1).

From these data it appears that C2 follicle architecture is highly stereotyped with little quantitative variance. All of our major conclusions apply to all three datasets. We agree that a detailed analysis of the C2 follicle architecture variance in ten or more samples is desirable, but this was not the objective of our study.

Interpretive issues:

1. Gerhardt et al should clarify the issue of angular coverage stated in "We conclude that the bulk afferent innervation is polarized to posterior angles and that Merkel-afferents evenly tile the vibrissa follicle circumference." Our reading is that the Merkel cells are uniformly distributed over angles (Fig. 4e) with complete if nonuniform innervation (Figs. 3b, 4d). If this is correct, why do the authors discuss a preference for contact upon protraction? Is it because the club endings appear to have a bias in angular distribution (Fig. 3c)? Further to the point, if the ringwulst is a constant volume structure, then, gap aside, compressing it on one side will expand it on the other and stretch sensory endings.

Comment: The referee discusses the polarization of different afferent subtypes and how all findings pertain to this question. We have two comments here:

1. Our innervation polarization is the sum of angular distribution of all afferent types. Mostly Ruffini and lanceolate contribute to this effect and Merkel afferents contribute slightly to this effect (see Ref. Figure 1 below). It is suggested that angular preference of afferents overlaps with their angular position for most afferent types (Furuta, Harmann and colleagues, Current Biology 2020).
2. We do not know for certain, how afferents activation relates to the ringwulst. In particular, Furuta, Hartmann et al., and Thompson (PhD thesis) described a fair bit of direction specificity in club like afferent responses. We are therefore not sure if the ringwulst is in all cases activated as a rigid structure.

Change: We discuss our findings of dorso-caudal polarization, which aligns with preferences for upward deflections in the brainstem (see line 309 - 312).

**Referee Figure 1: Angular distribution by afferent type.**

2. Gerhardt et al finds a cluster of deep, unmyelinated axons that arrive at the follicle via the deep vibrissal nerve. These neurons preferentially innervate the posterior aspect of the whisker shaft with a tight distribution (Fig. 4d) at the top of the cavernous sinus. These axons are somewhat mysterious to me. Detailed GSA and IHC work of the rat follicle (Comprehensive

immunofluorescence and lectin binding analysis of vibrissal follicle ... JCN 1998) showed that the densest innervation site of unmyelinated axons was the inner conical body, distal to this site. However, there also should be unmyelinated axons present along the length of the follicle. These axons are generally described as being widely distributed. The authors need to reconcile this past claim with the somewhat precisely localization of axons in their study. The authors note that “reconstruction of unmyelinated axons was hampered by resolution limitations and is partially incomplete” Given these limitations, contradictions with current literature of unmyelinated axons in the follicle, and one sample, it seems best to exclude this finding of asymmetry from the analysis.

Comment: The referee describes several of our findings on unmyelinated axons and contrasts them with previous findings from the literature. We have several comments here:

1. The assessment of thin unmyelinated axons is not a strength of our analysis, because we meet resolution limits here.
2. We are confident that there is a massive asymmetric innervation of ventral follicle regions by unmyelinated axons.
3. Because of resolution limitations we might have missed unmyelinated axons elsewhere in the follicle and in the inner conical body.
4. A sizeable proportion of axons innervating the inner conical body was myelinated in our sample.
5. We now cite the study referred to by the referee (Rice et al., JCN 1997, ref 38), when discussing our findings on unmyelinated axons (line 120 – 122 and 276 - 278). This paper assumes the superficial nerve as origin for inner conical unmyelinated axons, while the unmyelinated innervation shown in our manuscript is supplied by the deep vibrissal nerve. Hence there might be no contradiction with the reference mentioned.

Change: Given the resolution limitations of our study we emphasized the need to interpret the data referring to unmyelinated axons with caution in the result section (see line 121 - 123). We also added ref. Rice et al., JCN 1997 (ref 38). We also raise this issue in the discussion of our manuscript (line 277 - 279). Following the referee’s criticism, we now present polarization data of myelinated and unmyelinated fibers separately (newly added panel Figure 5e).

3. Gerhardt et al state that despite the angular distribution of sensory endings at the receptor level of the follicle (Fig. 5e), axons arranged themselves to represent angles linearly rather than radially when they leave the follicle complex (Fig. 5c,f). The authors then speculate this may be the mechanism for achieving linear somatotopy along the log-like structure of the

barreloids in thalamus. They also hypothesize that perhaps the unknown receptor directional tuning somatotopy of the trigeminal nuclei barrelettes would be linear by extension.

There is no statistical quantification for the claim of linearization. Further, the data is presented with a color-coding that seems to best magnify “linear” somatotopy by grouping single afferents according to their respective “axon arms”, or subcollection of neighboring receptors that cluster. Thus while the data in Fig 5c illustrates that there is overlap between the individual receptor locations in each “axon arm”, that of Fig. 5f removes this variability. In fact, it appears that the upper fascicles in Fig. 5f codes contact to the caudal part of the follicle, as in contact during protractions, while the lower fascicle codes contact during retraction.

Comment: We have three comments here:

1. We agree with the referee that our manuscript lacked quantitative rigor to support afferents linearization. We addressed this issue by additional analysis (see below).
2. A second criticism of the referee is that the visualization (i.e. color coding of afferents by arm identity) in Figure 5 (now Figure 6) might obscure correct assessment of actual nerve topography by occluding angular outliers. We address this criticism with the Referee Figure shown below.
3. The referee suggests that the two different sub-fascicles of the deep vibrissal nerve might receive afferents related to protraction and retraction contacts respectively. We carefully studied which afferents angular positions sort into the respective sub-fascicles, but our analysis does not support the idea of protraction-retraction sorting of afferents. Instead, it appears, that afferents are sorted roughly but not precisely according to an axis orthogonal to the ringwulst-aperture-vibrissal-shaft-direction. As the ringwulst aperture points dorso-caudally, we do not expect a sorting of afferents according to protraction-retraction.

Change:

1. We performed additional analysis to rigorously quantify afferents linearization in the newly added Figure 6g, 6h subpanels. We find that a afferents sequence in the nerve indeed linearly relates to the angular position of afferent terminals.
2. We added Referee Figure 2 which shows that the linear nerve layout is not an artifact of color coding by axon-arm identity. Specifically, when we color code merely by angular position (panel a), the same linearization is observed in the nerve crosssection as revealed by axon-arm coding (panel b).
3. No change.

Referee Figure 2: Linear nerve layouts both by arms and terminal angular positions.

a, Continuous color coding of afferents by terminal angles around the vibrissal shaft, as suggested by the referee (left) leads to a similar, banded appearance of the nerve layout (right).
b, Discrete color coding by axon-arm position around the vibrissal shaft (as in Figure 6) shows a similar pattern of nerve layout as in a.

4. Gerhardt et al should comment on how axial forces generated by whisker contact will be detected and encoded by the different receptor groups.

Comment: We agree.

Change: We added a paragraph on this in the discussion (see line 292 – 302).

Minor issues:

1. The synopsis "Afferent responses are tuned to the angular displacement of the shaft, respond with astounding temporal precision providing information about object distance and orientation." should include reference to the awake animal results of Ref. 29, which shows modulation of trigeminal spiking based on mystacial muscle activity and on phase in the whisking cycle.

Comment: We agree that this reference should have been mentioned there.

Change: We added the reference in question.

2. The statement "Vibrissa retraction results from tissue elasticity and a smaller set of so-called extrinsic whisker muscles anchored outside of the whisker pad (not visible in Fig. 4b)." should include Ref. 31.

Comment: We agree that this reference should have mentioned there.

Change: We added the reference in question.

3. The reference following "... the contraction of which leads to vibrissa protraction" should include Simony et al (J Neurosci 2010), co-authored by Brecht, in addition to Refs. 29 and 30.

Comment & Change: We agree and implemented this change.

4. Please clarify "Characteristic shaft bending moments provide pre-neural information of object touch locations". The bending is a measure of the applied torque, and the associated force will have lateral and axial components. Where is the "pre-neural information" encoded?

Comment: We apologize for being unclear in our citation.

Change: We changed the wording for improved clarity.

5. I appreciate the inability to reconstruct all unmyelinated axons; the IARPA MICrONS EM project has the same issue. Is there a sense that this could affect the conclusion on the anisotropy of innervation (Fig. 4d).

Comment: We agree, please see our response to major point 2.

Change: Please see our response above.

6. The attachment to the intrinsic muscles is shown (Fig. 4b). Yet attachment to the skin and plate (in the notation of Dorfl, Ref. 30) is not determined. This systematic issue limits the mapping of the data to a mechanical model.

Comment: The criticism of the referee is justified and is the same as the critique of referee 1 (point 12).

Change: We improved our microCT reconstruction of muscles and added the missing detail.

Bottom Line:

There are some exciting claims in this work. Unfortunately the reliability of the claims is overshadowed by the lack of replicates and thus the inability to distinguish variability between animals as opposed to purposeful anatomical organization.

Comment & Change: We think this bottom-line summary is somewhat too harsh:

1. Our dense reconstruction of vibrissal afferents is a first of its kind.
2. In the revised manuscript we show three stunningly similar completely reconstructed and analyzed datasets (Supplementary Figure 1). It is clear from these data that the variability of C2 vibrissa follicle architecture is low and that all our key conclusions apply to all datasets (Supplementary Table 1). Hence, it appears the referee's criticism does not hold.

***We are aware that Brecht successfully used x-ray CT to look at muscles (Kaufmann et al, Sci Adv 2022; Longren et al, Curr Biol 2023) and innervation (Purkart et al, Curr Biol 2022) of the trunk of the baby elephant in a series of "N = 1" studies. Yet those were pioneering one-off experiments that further relied on the dearth of elephants compared to rats.*

Comment: We agree.

Change: None.

Reviewer #3 (Remarks to the Author):

Comment: We appreciate the inclusion of junior reviewers. We feel we got very detailed and very substantial feedback on our manuscript, for which we are grateful.

Change: None.

**Three-dimensional architecture and linearized mapping of vibrissa follicle afferents
- Revision round 2 -**

Dear editors and referees,

We are most thankful for the support of publishing our manuscript. We answer the remaining questions below and implemented changes in the manuscript as specified.

Reviewer #1 (Remarks to the Author):

Thanks again for sharing with me this wonderful paper – it is pure joy:)

The authors have satisfactorily addressed all my previous concerns and exceeded expectations, producing exciting new results.

Upon re-reading, I have a few additional comments that may help refine and clarify their key messages.

Comment: We are deeply thankful for the referees positive assessment of our manuscript and the compliments on our revision effort.

Minor:

Proximal-to-distal gradient in axon diameter. Examination of fig.2f, S2c and S4a suggests to me that instead of a gradient, there are two modes of diameters – small, for < 850 microns, and large for >850 microns. I don't see a gradient in the latter. I suggest that the authors will reconsider the classification here. The difference between a gradient and 2-modes may become important for future evaluations of the evolutionary steps involved in follicle specialization.

Comment & Change: We agree with the referee that afferent diameter appear to follow a bimodal distribution above and below the 850 μm segment above nerve entrance into the follicle. We now further highlight this finding by adding the distribution plot as a panel in the main figure 2h and referring to two modes of fiber diameter in the related results paragraph.

Fig. 3e – add in legend that black stands for the presence and white for the lack of a property. In x-axis label write “Axon count” or something alike, rather than “Axon”.

Comment & Change: We changed the x-axis labelling as suggested.

Do individual arms contain a mix of the two types of club-like fibers or not? In addition, and in a related manner, could the authors speculate regarding the evolutionary order of the two club-like types?

Comment & Change: Both club-like subtypes co-occur in the same axon arms. We added a sentence on this in the related results paragraph. While our data provides no direct insights into evolutionary order of club like afferents and their subtypes, we may only speculate about this. We suggest, that the club like receptor type evolved from lanceolate endings and that the large caliber subtype evolved first, given the similarity to lanceolate endings, which we speculate to be the evolutionary origin of the club like receptor type.

Reviewer #2 (Remarks to the Author):

I support publication in light of the answers to my queries.

Comment & Change: We are glad that we could sufficiently answer questions of the referee and are thankful for the support of publication.

Reviewer #3 (Remarks to the Author):

Comment & Change: We thank the referee for co-reviewing our manuscript. We think that we received helpful feedback, which improved the manuscript.